# Hemispherically-Symmetric Strategies for Stratospheric Aerosol Injection

Yan Zhang[1], Douglas G. MacMartin[1], Daniele Visioni[2], Ewa Bednarz[1,3,4], and Ben Kravitz[5,6]

[1]Sibley School of Mechanical and Aerospace Engineering, Cornell University, Ithaca, NY, USA
[2]Department of Earth and Atmospheric Sciences, Cornell University, Ithaca, NY, USA
[3]CIRES, University of Colorado Boulder, Boulder, CO, USA
[4]NOAA Chemical Sciences Laboratory, Boulder, CO, USA
[5]Department of Earth and Atmospheric Science, Indiana University, Bloomington, IN, USA
[6]Atmospheric Sciences and Global Change Division, Pacific Northwest National Laboratory, Richland, WA, USA

**Correspondence:** Yan Zhang (yz2545@cornell.edu)

**Abstract.** Stratospheric aerosol injection (SAI) comes with a wide range of possible design choices, such as the location and timing of the injection. Different stratospheric aerosol injection strategies can yield different climate responses; therefore, understanding the range of possible climate outcomes is crucial to making informed future decisions on SAI, along with the consideration of other factors. Yet to date, there has been no systematic exploration of a broad range of SAI strategies. This limits the ability to determine which effects are robust across different strategies and which depend on specific injection choices. This study systematically explores how the choice of SAI strategy affects climate responses in one climate model. Here, we introduce four hemispherically-symmetric injection strategies, all of which are designed to maintain the same global mean surface temperature: an annual injection at the equator (EQ), an annual injection of equal amounts of $SO_2$ at $15°$ N and $15°$ S (15N+15S), an annual injection of equal amounts of $SO_2$ at $30°$ N and $30°$ S (30N+30S), and a polar injection strategy that injects equal amounts of $SO_2$ at $60°$ N and $60°$ S only during spring in each hemisphere (60N+60S). We compare these four hemispherically-symmetric SAI strategies with a more complex injection strategy that injects different quantities of $SO_2$ at $30°$ N, $15°$ N, $15°$ S, and $30°$ S in order to maintain not only the global mean surface temperature but also its large scale horizontal gradients. All five strategies are simulated using the Earth system model CESM2(WACCM6-MA), with the global warming scenario SSP2-4.5. We find that the choice of SAI strategy affects the spatial distribution of aerosol optical depths, injection efficiency, and various surface climate responses. In addition, injecting in the subtropics produces more global cooling per unit injection, with the EQ and the 60N+60S cases requiring, respectively, 59 % and 50 % more injection than the 30N+30S case to meet the same global mean temperature target. Injecting at higher latitudes results in larger equator-to-pole temperature gradients. While all five strategies restore September Arctic sea ice, the high-latitude injection strategy is more effective due to the SAI-induced cooling occurring preferentially at higher latitudes. These results suggest trade-offs wherein different strategies appear better or worse depending on what metrics are deemed important.

# 1 Introduction

Current climate projections suggest that under most emission scenarios the $1.5°C$ threshold of global mean temperature increase above pre-industrial levels set by the Paris Agreement is likely to be exceeded by 2040 or earlier (IPCC, 2021; Tebaldi et al., 2021; Dvorak et al., 2022). Meinshausen et al. (2022) showed that implementing all conditional and unconditional Paris Agreement pledges on time may limit global warming to just below $2°C$. With the uncertainties in the implementation of carbon emission reductions, estimates of climate sensitivity, and severity of impacts of climate change, only relying on carbon emission reduction is likely insufficient to reduce the possibility of severe adverse climate impacts in the foreseeable future (Rogelj et al., 2016; Bamber et al., 2019; Anderson et al., 2020; Sherwood et al., 2020; Bjordal et al., 2020; MacMartin et al., 2022). This leads to the suggestion that stratospheric aerosol injection (SAI) could be an option at some point to reduce severe adverse impacts on climate and society. Such an approach would consist of injecting aerosols, or their precursors, in the lower stratosphere to reflect a small fraction of the incoming solar radiation back to space, as a result, lowering the global mean temperature. In this study, we focus only on $SO_2$ injections.

To inform future decisions on SAI deployment, it is important to have a sufficient understanding of the range of possible climate responses under SAI; these would depend on both the scenario and strategy. However, most existing SAI studies looking at surface impacts consider only a single scenario (i.e. a particular choice of background emission scenario, deployment start date and desired temperature target to be achieved with SAI) and only look at a single SAI strategy (i.e. a particular choice of injection latitude(s) and season(s)) (Kravitz et al., 2019; Visioni et al., 2020b; Tilmes et al., 2018; Irvine et al., 2019). Recently, MacMartin et al. (2022) and Visioni et al. (2023b) explore a set of specific scenario choices that cover a range of plausible futures, all with a single strategy. Here we consider and compare a set of different SAI strategies under the same scenario. Collectively, MacMartin et al. (2022) and this study capture two key dimensions of the range of possible climate responses to SAI.

Different SAI strategies could result in the same level of global cooling, but affect the regional surface climate differently (Visioni et al., 2020b; Kravitz et al., 2019; Lee et al., 2020, 2021; Zhang et al., 2022). Injecting $SO_2$ at the equator would overcool the tropical region and undercool the high-latitude regions (Kravitz et al., 2019); this was a key motivation in developing a multi-objective strategy in Kravitz et al. (2017) that injects at multiple latitudes to balance not just global mean temperature, but also interhemispheric and equator-to-pole temperature gradients. This multi-objective strategy was used in the Geoengineering Large Ensemble Project (GLENS; Tilmes et al., 2018) and more recent studies (MacMartin et al., 2022; Richter et al., 2022). Injecting at $60°N$ would primarily cool the Northern Hemisphere (Lee et al., 2023a). Injecting $SO_2$ in the same latitude but in different seasons may also result in slightly different regional climate responses (Visioni et al., 2020b). Knowing the dependence of various climate responses on the choice of SAI strategies is crucial for comparing the benefits and risks of different SAI strategies. In addition, SAI will not bring the climate back to the same state as lowering the $CO_2$ concentration; instead, it will create a novel climate (Bala et al., 2010; Niemeier and Timmreck, 2015; Kravitz et al., 2017; Tilmes et al., 2018; Irvine et al., 2019). Knowing the range of possible climates and how close we can bring the climate to a reference state by SAI will enable us to evaluate the limits of SAI and the trade-offs between achieving different climate goals.

In this study, we simulate four hemispherically-symmetric injection strategies in order to explore the range of possible climate responses. These four strategies are annual injection of $SO_2$ at the equator (EQ), annual injection of equal amounts of $SO_2$ at $15°\,N$ and $15°\,S$ (15N+15S), annual injection of equal amounts of $SO_2$ at $30°\,N$ and $30°\,S$ (30N+30S), and spring injection of equal amounts of $SO_2$ at $60°\,N$ and $60°\,S$ (60N+60S; it is referred to as POLAR in Bednarz et al. (2023a) and Polar+1.0 in Goddard et al. (2023)), all designed to maintain a targeted global mean temperature. We assess a broad range

of differences between these strategies to illustrate trade-offs between them; this understanding can lay the foundation for future work to develop "better" strategies as well as motivate the design of future multi-model intercomparisons. Section 2 describes the climate model used herein. Section 3 explains how this set of strategies is chosen and describes the simulation setup. Section 4 describes the simulation results of the four new strategies and compares them to the multi-objective strategy developed in Kravitz et al. (2017) and simulated in MacMartin et al. (2022) (see also the companion paper Bednarz et al.

(2023a) that compares the stratospheric response for these strategies).

## 2    Climate Model

All SAI strategies are simulated using version 2 of the Community Earth System Model with the middle atmosphere version of the Whole Atmosphere Community Climate model, version 6, as the atmospheric component, CESM2(WACCM6-MA) (Danabasoglu et al., 2020; Gettelman et al., 2019; Davis et al., 2023). CESM2(WACCM6-MA) is a fully coupled Earth system

model which includes atmosphere, ocean, land, and sea ice components. The middle atmosphere (MA) version of WACCM6 uses chemistry mechanisms relevant for the stratosphere and mesosphere with a reduced set of tropospheric reactions (Davis et al., 2023), similar to the chemistry configuration in CESM1(WACCM). The ocean component is based on the Parallel Ocean Program Version 2 (POP2), the land component is Community Land Model Version 5 (CLM5), and the sea ice component is CICE5 (Danabasoglu et al., 2020). The horizontal resolution of CESM2(WACCM6-MA) is $0.95°$ in latitude and $1.25°$ in lon-

gitude, with 70 vertical layers extending from the Earth's surface to about $140\,\mathrm{km}$ in altitude, the same as in CESM1(WACCM) (Mills et al., 2017). The stratospheric aerosol distribution in this model reasonably matches observations after the 1991 eruption of Mt. Pinatubo (Mills et al., 2017).

## 3    Simulations

While different SAI strategies would not result in the same surface climate, the differences in surface climate responses be-

tween some SAI strategies would be much easier to detect than between others. The detectability of the differences in surface climate responses between SAI strategies depends on, among other factors, the level of global cooling and natural variability. Zhang et al. (2022) estimated based on Community Earth System Model (CESM1) simulations that for a SAI-induced global cooling of $1$–$1.5°\,C$, there are only 6–8 injection choices that would produce detectably different surface climate responses, where two injection choices are considered detectably different if the difference in temperature or precipitation responses are

detectable at a 95% confidence level over a 20-year period on more than 5% of the Earth's area. Although the estimate of

6–8 distinct injection choices was made using CESM1(WACCM) simulations, the conclusion is expected to hold relatively well in CESM2(WACCM) due to similarities in the stratospheric circulation and aerosol microphysics between the two model versions. This is demonstrated by the results of a set of fixed-amount single-latitude injection simulations (Fig. S1 in supplementary material). For a global cooling level of 1–1.5°C, a reasonable choice of seven latitudes of injection with patterns of AOD that would yield detectably different surface climate responses is: 60°N, 30°N, 15°N, the equator, 15°S, 30°S and 60°S (Zhang et al., 2022). These seven latitudes could be combined in different ways to form a set of seven linearly independent injection strategies that span the same AOD design space. The outcomes of other strategies can be estimated by a linear combination of these seven injection strategies, assuming linearity (MacMartin et al., 2017, 2019; Zhang et al., 2022).

Here, we simulate and compare four hemispherically-symmetric injection strategies that collectively cover the seven latitudes mentioned above, with the consideration of ensuring hemispheric equality in the deployment of SAI. These four injection strategies are: injecting solely at the equator (EQ), injecting the same amount at 15°N and 15°S (15N+15S), injecting the same amount at 30°N and 30°S (30N+30S), and injecting the same amount at 60°N and 60°S in springtime only in each hemisphere (60N+60S) (Table 1). These new strategies are designed to maintain the same global mean surface temperature ($T_0$). The global mean surface temperature is the metric used by the United Nations Framework Convention on Climate Change (UNFCCC) to operationalize climate change goals in the Paris Agreement (UNFCCC, 2015), and is thus a reasonable metric to consider as a target for SAI (MacMartin et al., 2022). In addition, we compare the new strategies simulated herein with a multi-objective strategy simulated in MacMartin et al. (2022) that maintains not only $T_0$, but also the interhemispheric temperature gradient ($T_1$) and equator-to-pole temperature gradient ($T_2$).

The multi-objective strategy adjusts the $SO_2$ injection rates at 30°N, 15°N, 15°S, and 30°S to maintain $T_0$, $T_1$ and $T_2$. Managing the interhemispheric temperature gradient is motivated by the desire to reduce shifts in tropical precipitation; however, the specific injection rates have been shown to vary even in different versions of the same Earth System Model (Fasullo and Richter, 2023). While the radiative forcing from $CO_2$ is roughly hemispherically symmetric, other effects such as rapid cloud responses to elevated $CO_2$ levels and changes in Atlantic meridional overturning circulation (AMOC), lead to changes in $T_1$ that require asymmetric injection to compensate. These effects are model dependent; for example, in CESM1(WACCM), more injection is needed in the Northern Hemisphere (NH) to compensate $T_1$, but in CESM2(WACCM6) more injection is needed in the Southern Hemisphere (SH) (Fasullo and Richter, 2023). Because the sign of the hemispheric asymmetry in injection rates that is needed to maintain $T_1$ varies among different climate models, here we focus on hemispherically symmetric strategies that maintain only $T_0$ (Table 1). While we do not expect these to fully balance the interhemispheric temperature gradient $T_1$ in CESM2(WACCM6), these strategies are simpler to implement in other climate models, as the injection rate could be adjusted to meet the only objective ($T_0$) by hand. Simultaneously tuning multiple variables is more challenging without explicitly coding a feedback control algorithm.

In addition to the multi-objective strategy and the four hemispherically-symmetric strategies, a complete set of strategies spanning the space of the seven injection choices described by Zhang et al. (2022) would also include two other strategies, such as a spring injection at 60°N (Lee et al., 2023a) and an annually-constant injection at 30°N (Bednarz et al., 2022b). However, injecting outside of the tropics but in a single hemisphere would primarily cool that hemisphere, which would result in a

significant perturbation of the interhemispheric temperature gradient and the associated location of tropical precipitation (Haywood et al., 2013). Thus, these or any other extratropical single-latitude injections are already known to not be an appropriate strategy for targeting global mean temperature and as such are not included in the analysis discussed here.

All of the strategies considered herein are simulated under the same scenario (i.e., the same background greenhouse gas emissions, start date for SAI deployment, and global mean temperature target). The background emission scenario used here is the Shared Socioeconomic Pathway (SSP) 2-4.5 (Meinshausen et al., 2020), a 'middle-of-the-road' pathway in which the world is facing medium challenges to mitigation and adaptation (IPCC, 2021). This background emission scenario is roughly consistent with the Paris Agreement's Nationally Determined Contributions (Burgess et al., 2021; UNEP, 2021-10). All of these injection strategies are simulated from the beginning of 2035 to the end of 2069. The average over 2020–2039 in the model is chosen to be representative of when future climate might reach $1.5°\,$C above pre-industrial levels (MacMartin et al., 2022). Here, to increase the ability to distinguish between effects of different strategies, we choose an additional $0.5°\,$C cooling relative to the $1.5°\,$C target from the Paris Agreement. This new temperature target of $1.0°\,$C above pre-industrial levels corresponds to the average global mean temperature over 2008–2027 in CESM2(WACCM6), which we will use as the reference period for comparison. All simulations herein aim to ultimately cool the planet to this $1.0°\,$C target, but as the model temperature in 2035 (i.e. at the start of SAI deployment) is already roughly at $1.5°\,$C above pre-industrial levels, the cooling target gradually ramps down to the desired $1.0°\,$C target over the first 10 years of simulation and then stays the same for the following years. This corresponds to the SSP2-4.5:1.0 scenario in MacMartin et al. (2022).

We choose the injection altitude as $21.5\,$km for injection latitudes from 30°N to 30°S as in MacMartin et al. (2022), consistent with plausible estimates of engineering feasibility, and choose $15\,$km for injecting at 60°N and 60°S, where the tropopause is lower, as in Lee et al. (2023a). The altitude of injection will affect the aerosol lifetime and thus the injection rate needed to achieve a desired cooling (Lee et al., 2023b).

Injection rates are determined by a controller, which has a feedforward component and a feedback component. At the start of each model year, the controller takes the output values from the previous year and calculates the injection rate for the forthcoming year. The feedforward component estimates the required global mean AOD based on a simple quasi-static linear model, using the rate of warming in the SSP2-4.5 scenario ($0.0273\ °$ C yr$^{-1}$) and the sensitivity of global mean temperature to global mean AOD, $\frac{\Delta T_0}{\ell_0}$. The sensitivity of global mean temperature to global mean AOD is estimated from 10-year single-latitude fixed injection-rate simulations in Visioni et al. (2023a), giving 3.9, 4.4, 5.4, and $8.3°\,$C for EQ, 15N+15S, 30N+30S and 60N+60S, respectively. In the feedback component, a Proportional Integral (PI) controller is designed to correct the estimated global mean AOD based on the measured difference between actual and reference values of global mean temperature from the previous model year (MacMartin et al., 2014; Kravitz et al., 2017):

$$\ell_{0_{t+1}} = \hat{\ell}_{0_{t+1}} + k_p(T_t - T_{ref}) + k_i \sum_{j=1}^{t}(T_j - T_{ref}) \tag{1}$$

where $T_t$ denotes the global mean temperature in the year of simulation that was just completed, $T_{ref}$ denotes the targeted global mean temperature, and $\hat{\ell}_{0_{t+1}}$ and $\ell_{0_{t+1}}$ denote the estimated global mean AOD that is needed to compensate for the global mean temperature in the forthcoming year before and after correction by the feedback algorithm. The $k_p$ and $k_i$ are the

**Table 1.** SAI Strategies evaluated in this study. All simulations start in January 2035 and end in December 2069. Spring season is March, April, and May (MAM) for the Northern Hemisphere, and September, October, and November (SON) for the Southern Hemisphere.

| Strategy | Injection rate and latitude(s) | Injection season | Injection altitude (km) | Design objective(s) |
|---|---|---|---|---|
| 60N+60S | equal amounts at $60^\circ$ N and $60^\circ$ S | Spring (MAM at $60^\circ$ N, SON at $60^\circ$ S) | 15.0 | $T_0$ |
| 30N+30S | equal amounts at $30^\circ$ N and $30^\circ$ S | Annually constant | 21.5 | $T_0$ |
| 15N+15S | equal amounts at $15^\circ$ N and $15^\circ$ S | Annually constant | 21.5 | $T_0$ |
| EQ | equator | Annually constant | 21.5 | $T_0$ |
| Multi-Objective (MacMartin et al., 2022) | different amounts at $30^\circ$ N, $15^\circ$ N, $15^\circ$ S, and $30^\circ$ S | Annually constant | 21.5 | $T_0, T_1, T_2$ |

proportional and integral gains. These are set to be equal in the PI controller for each SAI strategy as described in Kravitz et al. (2016, 2017) and are scaled from the values used in MacMartin et al. (2022) based on the relative sensitivity of temperature to AOD obtained from the 10-year simulations in Visioni et al. (2023a), giving 0.0206, 0.0183, 0.0149, and 0.0097 for EQ, 15N+15S, 30N+30S and 60N+60S, respectively. With the estimated global mean AOD required to meet the desired temperature target, the injection rates for the forthcoming year are chosen based on $q_{t+1} = \alpha \ell_{0_{t+1}}$, where $\alpha$ is again estimated from 10-year
simulations, as 59.30, 60.00, 63.77, and 117.66 $\mathrm{Tg\ yr}^{-1}$ for EQ, 15N+15S, 30N+30S and 60N+60S, respectively.

The SSP2-4.5 and all SAI cases consist of three ensemble members each. The surface climate responses are evaluated based on the 20-year average over the period of 2050–2069. With three ensemble members, the 20-year average of each evaluated climate variable is calculated based on 60 annual-mean values. Taking into account temporal autocorrelation, the effective sample size is still comparable to the suggested number of independent data points (20-40) in Pausata et al. (2015). This
effective sample size is also comparable to the suggested sample size (7-40) in another relevant study focusing on discerning NH polar vortex change from internal variability (Bittner et al., 2016). As this study focuses on the long-term impacts of continuous injection, rather than impacts of a pulse volcanic eruption in the single year following the eruption (as in, e.g., Pausata et al., 2015; Bittner et al., 2016), data from three ensemble members are likely sufficient to distinguish a signal over a 20-year period from internal variability.

 **4   Results**

Here we present the injection rates and stratospheric AOD values, as well as global and regional surface climate responses under the four hemispherically-symmetric SAI strategies and the multi-objective strategy. All of these five injection strategies are designed to maintain the same global mean surface temperature. The evaluated climate responses to these five strategies are estimated based on the annual-mean values over the period of 2050-2069; thus, 60 data points are collected for each evaluated climate variable under each injection strategy. When calculating statistical significance in Figures 8-10 and Figures S5-S6 in the supplementary material, we adjust the degrees of freedom to account for temporal autocorrelation, where we assume that a first-order autoregressive (AR(1)) model is an adequate approximation to estimate the effective sample size (Wilks, 2019). We also perform multiple testing correction to account for spatial correlation using the false discovery rate (FDR) method, where we choose $\alpha_{FDR} = 0.1$ for achieving a global significance level of 0.05 based on the conclusion in Wilks (2016). We use t-tests to estimate significance, which assume that variability is approximately normal; this is a reasonable approximation for annual-mean climate variables.

**4.1   Large-scale global climate responses**

Figure 1(a) shows the time evolution of the global mean surface temperature in all simulations. In the last 20 years of injection, $T_0$ in all SAI strategies considered here is maintained within one standard deviation ($\sigma_{T_0}$=0.24°C) from the target value; this corresponds to approximately 1.4°C global cooling compared to the SSP2-4.5 case without SAI. As discussed in Section 2, the multi-objective strategy is the only SAI strategy discussed here that is also designed to maintain the interhemispheric temperature gradient ($T_1$) and the equator-to-pole temperature gradient ($T_2$) in addition to $T_0$. $T_1$ and $T_2$ are defined as the linear and quadratic meridional dependence of the zonal-mean temperature (Kravitz et al., 2016):

$$T_1 = \frac{1}{A} \int_{-\pi/2}^{\pi/2} T(\psi) sin(\psi) \, \mathrm{d}A \tag{2}$$

$$T_2 = \frac{1}{A} \int_{-\pi/2}^{\pi/2} T(\psi) \frac{1}{2}(3 sin^2(\psi) - 1) \, \mathrm{d}A \tag{3}$$

where $\psi$ is the latitude in radians, $T(\psi)$ is the zonal-mean temperature at latitude $\psi$ and $A$ is the surface area of the Earth. A positive value of $T_1$ means that the Northern Hemisphere (NH) is warmer than the Southern Hemisphere (SH). $T_2$ is always negative because the polar regions are colder than the tropics; an increase in the temperature difference between the equator and poles will decrease $T_2$.

Without SAI, $T_1$ increases over time under climate change (Fig. 1(b)) due to various reasons, such as differences in land cover, tropospheric aerosol and heat capacity between the two hemispheres (Chiang and Friedman, 2012). We find that all SAI strategies considered here overcompensate $T_1$, which corresponds to a reduction in temperature gradient between the NH and SH compared to the reference period (this includes the multi-objective case that targets T1, although that case has the smallest overcompensation). The overcompensation of $T_1$ is likely linked to the reduction in cloud cover in the SH subtropics

due to the strong cloud response to elevated $CO_2$ levels in the SH in CESM2(WACCM6) (Fasullo and Richter, 2023). As a result, greater radiative heating needs to be mitigated in the SH. The same SAI strategies do not overcompensate $T_1$ in other models. For example, in CESM1(WACCM), the equatorial injection, which yields slightly larger AOD in the NH than the SH, roughly maintained $T_1$, as described in Kravitz et al. (2019). With greater radiative heating needed to be mitigated in the SH in CESM2(WACCM6) compared to CESM1(WACCM), the equatorial injection ends up overcompensating $T_1$ in this model.

Figure 1(c) shows the evolution of the equator-to-pole temperature gradient. $T_2$ increases over time under SSP2-4.5 as the result of the warming being much faster in the Arctic than in the mid- and low-latitudes. All SAI strategies considered here reduce $T_2$ compared to the SSP2-4.5 simulation. The strategies injecting further poleward i.e. 30N+30S and 60N+60S, overcompensate $T_2$ compared to the reference period, while the equatorial case undercompensates it. Some intuition for this is based on the observation that the radiative forcing from $CO_2$ is roughly uniform with latitude, while insolation is higher in the tropics than towards the poles. Thus, one would expect a spatially uniform AOD to overcool the tropics relative to high latitudes, overcompensating $T_2$. Injecting further poleward increases AOD further poleward, and in this model, injecting at 15N and 15S is roughly sufficient to balance the mismatch between the spatial distribution of radiative forcing from $CO_2$ and that of sunlight, and thus simultaneously balance $T_0$ and $T_2$ – essentially giving the latter for free while only directly controlling for $T_0$. A more complete description would depend on other factors, including details of the stratospheric circulation, and the rapid cloud adjustment to $CO_2$ forcing noted in Fasullo and Richter (2023); as a result, the specific injection latitudes that would simultaneously balance both $T_0$ and $T_2$ will be model dependent.

Figure 1(d) shows the evolution of global mean precipitation. With increasing GHG forcing, global mean precipitation increases over time in the SSP2-4.5 simulation. This response has been observed under rising GHG levels across climate models (IPCC, 2021), and arises because global mean precipitation is governed by the availability of energy (Allen and Ingram, 2002; O'Gorman et al., 2012). With the added SAI forcing, the global mean precipitation is reduced, consistent with the associated decrease in global mean temperature, and is overcompensated relative to the global mean precipitation in the reference period ($P_0$=2.9 mm day$^{-1}$), except for the 60N+60S case. This overcompensation in precipitation relative to the associated decrease in temperature was observed in many previous studies using either solar reduction (Bala et al., 2008; Tilmes et al., 2013) or stratospheric aerosols (Niemeier et al., 2013; Lee et al., 2020).

To understand what factors affect the overcompensation of global mean precipitation, we use the precipitation and temperature data from the five SAI strategies as well as SSP2-4.5 to calculate the hydrological sensitivity (the slope between global mean precipitation and global mean surface air temperature) under different SAI strategies. The results in Fig. 2 show that the hydrological sensitivity is dependent on the injection latitude; injecting $SO_2$ at lower latitudes yields a stronger reduction of global mean precipitation per unit of reduction in global mean temperature, as shown in Fig. 2(a). EQ has the strongest reduction in precipitation per unit of global cooling, followed by 15N+15S, multi-objective, and 30N+30S; the 60N+60S strategy has the least reduction in precipitation per unit of global cooling. This dependence on the injection latitude is also observed in the tropical region: injecting at lower latitudes yields a stronger reduction of tropical mean precipitation per unit of reduction in tropical mean temperature (Fig. 2(b)). It is likely that tropical cooling has a comparatively larger impact on global mean precipitation compared to the surface cooling that occurs outside the tropics, so the strategies with stronger tropical cooling

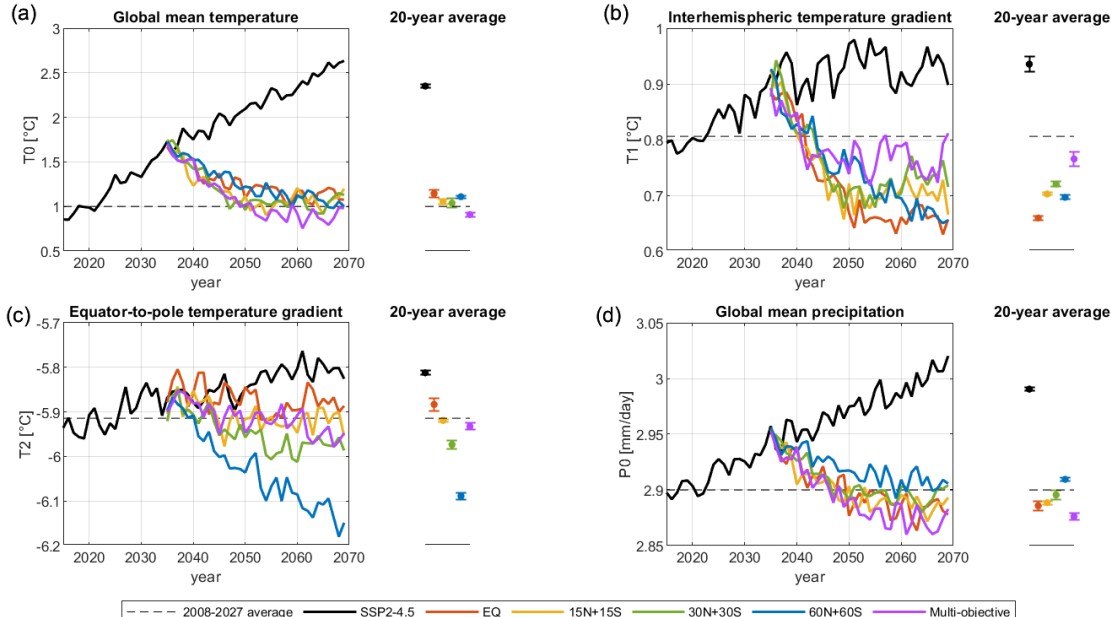

**Figure 1.** Time evolution of (a) global mean surface temperature relative to the pre-industrial level ($T_0$), (b) interhemispheric temperature gradient ($T_1$), (c) equator-to-pole temperature gradient ($T_2$), and (d) global mean precipitation ($P_0$). Each solid line represents the ensemble mean of each injection strategy. The dashed line represents the 20-year average during the reference period (2008–2027). The dots on the right of each panel represent the 20-year average over 2050–2069; the uncertainties in the calculated 20-year averages are estimated by $\pm 1$ standard error, and represented by the error bars.

yield stronger overcompensation in global mean precipitation (Fig. 2). In addition, the increase in tropospheric static stability as the result of aerosol-induced lower stratospheric heating can also contribute to the reduction of global mean precipitation (Simpson et al., 2019).

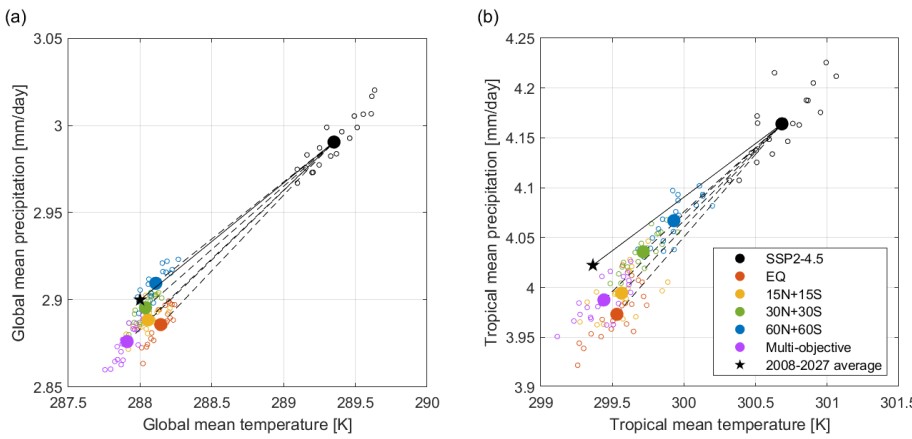

**Figure 2.** A comparison (a) between the global mean temperature and global mean precipitation, and (b) between the tropical mean temperature and tropical mean precipitation. The tropical means are calculated over the region between $20°$ N and $20°$ S. All data shown here are ensemble means. Small hollow dots represent the annual means from 2050–2069 under SSP2-4.5 or a given SAI strategy, and large solid dots represent the 20-year average over 2050–2069. The black star represents the 20-year average of temperature and precipitation from the reference period (2008–2027). The slope of the solid lines represents the increase in precipitation per unit of warming under GHG forcing. The slope of the dashed lines represents the precipitation reduction per unit of cooling under SAI forcing. The change in precipitation shows a strong dependence on injection latitude; injecting at lower latitudes yields a stronger reduction of global mean precipitation per unit of global cooling and a stronger reduction of tropical mean precipitation per unit of tropical cooling.

## 4.2 Injection rates and AOD

Figure 3 shows the evolution of the total $SO_2$ injection rate in each SAI strategy (Fig. 3(a)), and the 20-year (2050–2069)
average injection rates (Fig. 3(b)). Even though all five injection strategies aim to maintain the global mean surface temperatures at the same levels, different amounts of $SO_2$ injections are required in each case to achieve this. Among the five strategies, the 30N+30S strategy requires the least amount of injection, and the EQ and 60N+60S strategies require the largest amount of injection, which are, respectively, 59% and 50% more than the injection required by the 30N+30S strategy. The multi-objective strategy injects the majority (nearly 2/3) of the $SO_2$ in the Southern Hemisphere (Fig. 3(b)); the average injection rate during
2050–2069 at $30°$ S, $15°$ S, $15°$ N, and $30°$ N is 2.4, 8.8, 5.1, and 0.7 Tg yr$^{-1}$, respectively. This hemispheric asymmetry in the distribution of $SO_2$ injections is not due to Brewer-Dobson circulation as the interhemispheric imbalance for zonal mean AOD, $\ell_1$, under the hemispherically-symmetric strategies is much smaller than the value of $\ell_1$ that is needed by the multi-objective strategy to compensate for $T_1$. It is likely due to the rapid cloud responses to elevated $CO_2$ levels in CESM2(WACCM6), which results in greater radiative heating that needs to be mitigated in the SH (Fasullo and Richter, 2023).
The efficiency of AOD and of global mean surface cooling per unit injection for these five strategies is shown in Fig. 4(a) and (c), respectively. These results indicate that it is more efficient in terms of cooling per unit injection to inject $SO_2$ in mid-latitudes than in the tropics or high latitudes. The low efficiency in the equatorial injection is partially due to larger aerosol particles being formed near the tropics as the aerosols are relatively confined inside the tropical pipe and, hence, more prone

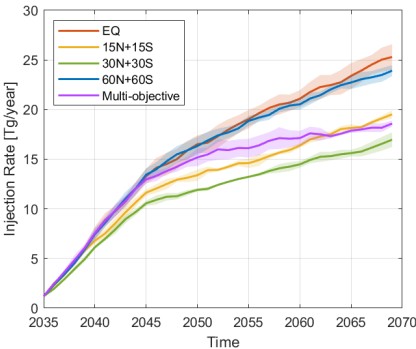 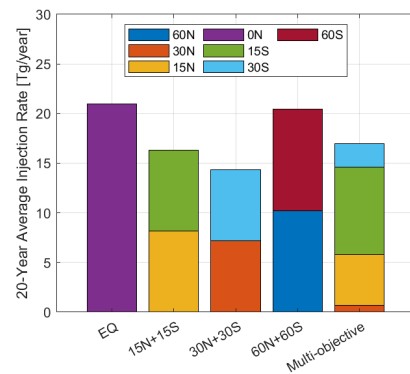

**Figure 3.** (a) Total amount of SO$_2$ injected into the stratosphere per year [Tg yr$^{-1}$], and (b) annual injection of SO$_2$ [Tg yr$^{-1}$] at each latitude averaged over the last 20 years (2050–2069), for each SAI strategy. The solid lines in (a) represent mean injection rate of each strategy, which is averaged over three ensemble members; the width of the shading represents the standard error of injection rates across ensemble members for each SAI strategy. The 20-year (2050–2069) average injection rates of EQ, 15N+15S, 30N+30S, 60N+60S, and multi-objective strategies are 21, 16, 14, 20, and 17 Tg yr$^{-1}$, respectively.

to coagulation and condensation (Fig. 5(b); see also Visioni et al. (2017); Kravitz et al. (2019)). The relatively larger aerosol
effective radius in the equatorial injection case notably reduces the AOD per unit mass of sulfate, and also slightly reduces the
aerosol lifetime in the stratosphere due to increased sedimentation. This results in the strongest increase in stratospheric water
vapor which, as a greenhouse gas, offsets some of the direct aerosol cooling (Visioni et al., 2021; Bednarz et al., 2022a); this
effect thus requires increased SO$_2$ injection rates to compensate.

    The notably lower efficiency of AOD per unit of injection in the 60N+60S strategy is because aerosols injected at high
latitudes have a much shorter lifetime due to the proximity to the downward branch of the stratospheric Brewer-Dobson
circulation and stratosphere-troposphere exchange areas, thus resulting in faster transport to the troposphere where they are
removed (Butchart, 2014; Lee et al., 2021; Visioni et al., 2023a). The average lifetime of the injected stratospheric aerosol
(calculated as the ratio of stratospheric SO$_2$ burden to injection rate) is 1.36±0.009 years, 1.39±0.011 years, 1.26±0.010
years and 0.58±0.004 year for the strategies EQ, 15N+15S, 30N+30S, and 60N+60S respectively. Although 60N+60S has the
lowest efficiency of AOD per unit injection, it yields the highest efficiency of global cooling per unit of global mean AOD
(Fig. 4(a)-(b), 5(a)), due to its strong effectiveness in offsetting Arctic amplification (Zhao et al. (2021); see also Section 4.1),
as the initial cooling from high latitude AOD is amplified by the high latitude feedbacks (Holland and Bitz, 2003; Serreze and
Barry, 2011; Hahn et al., 2021; Previdi et al., 2021). Figure 4(b) also indicates that the efficiency of global cooling per unit
AOD increases with latitude.

Nonlinearity is observed in the efficiency of AOD per unit injection, more notable in the low- and mid-latitude injections
(Fig. 5(a)). Higher concentration of SO$_2$ in the stratosphere results in larger aerosol particles which in turn sediment out faster,

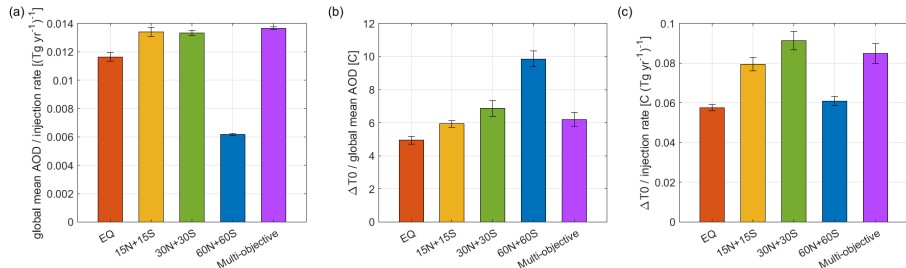

**Figure 4.** (a) Global mean AOD per unit of injection rate $[(\text{Tg yr}^{-1})^{-1}]$, (b) global cooling per unit of global mean AOD [C], and (c) global cooling per unit of injection rate $[\text{C}(\text{Tg yr}^{-1})^{-1}]$, calculated over the 20-year period of 2050–2069. Error bars represent the standard error of the mean.

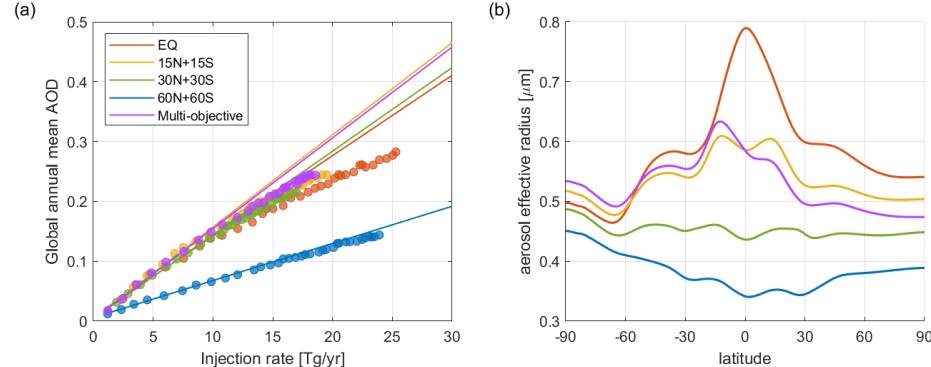

**Figure 5.** (a) The relationship between Injection rate and corresponding global mean AOD in each year of each simulation, and (b) latitudinal distribution of concentration-weighted aerosol effective radius in the stratosphere, averaged over the last 20 years (2050–2069). The lines in (a) are linear fits under low injection rates (i.e. when the injection rate is lower than 10 Tg yr$^{-1}$).

thus leading to smaller AOD per unit mass of sulfate (Niemeier and Timmreck, 2015; Kleinschmitt et al., 2018; Visioni et al., 2020a). Compared to high-latitude injection, low- and mid-latitude injections result in larger aerosol effective radius (Fig. 5(b)).

Figure 6 shows the latitudinal distributions of the zonal mean AOD and zonal mean temperature changes for different SAI strategies, averaged over the last 20 years of the simulations (2050–2069). Injecting in the tropics yields an asymmetrical AOD distribution between hemispheres, with higher AOD in the NH and lower AOD in the SH. This asymmetry arises as the Northern Hemisphere has a stronger Brewer-Dobson circulation than the Southern Hemisphere (Butchart, 2014). In contrast, injecting in the extratropics results in a relatively hemispherically-symmetric distribution of AOD. With the multi-objective strategy, AOD in the SH is notably higher than the NH, consistent with the largest injection rates at $15°$ S (Richter et al., 2022) that are required to minimize changes in the interhemispheric surface temperature gradient (Fig. 6(b)). Although the hemispherically-symmetric strategies yield similar levels of AOD at high latitudes in both hemispheres, the cooling in the Arctic is much larger than in the Antarctic (Fig. 6(b)-(c)) due to polar amplification asymmetry (Salzmann, 2017).

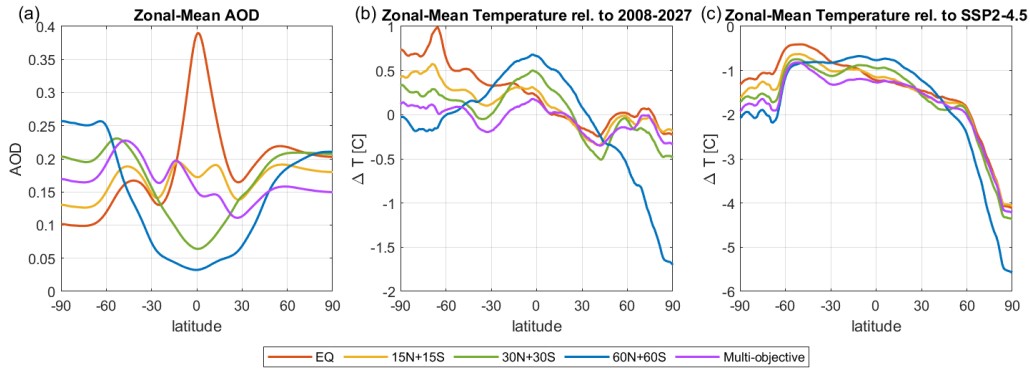

**Figure 6.** Latitudinal distribution of (a) zonal-mean AOD per degree Celsius of global cooling, (b) zonal-mean surface air temperature response relative to the 20-year average of the reference period, 2008–2027, and (c) zonal-mean surface air temperature response relative to the same 20-year period (2050–2069) under the SSP2-4.5 scenario.

Figure 7 shows the spatiotemporal distribution of stratospheric AOD for all five SAI strategies. We normalize the values of AOD by the associated amount of global mean cooling under each SAI strategy. The simulated distribution of AOD depends

on the latitudinal transport of air toward the poles, which is affected by both the seasonality in the Brewer-Dobson circulation and the strength of the stratospheric polar vortex (Visioni et al., 2020a). The distribution of AOD in the annual injection cases exhibits a marked seasonal cycle, with extratropical AOD maximizing in winter and spring at each hemisphere, due to seasonality in the strength of the stratospheric transport. In the case of the high-latitude seasonal injections, AOD maximizes in the mid- and high- latitudes in the season following the season of $SO_2$ injections because it takes about 1 month for injected

$SO_2$ to oxidize into aerosols (Lee et al., 2021).

### 4.3  Surface air temperature, precipitation, and P-E

Sections 4.1-4.2 above focused on the large-scale global responses to different SAI strategies; we now evaluate the corresponding changes in regional surface climate over the whole Earth surface. We average the annual mean surface air temperatures, precipitation and precipitation minus evaporation (P-E) over the 2050–2069 period and all three ensemble members, and cal-

culate the changes relative to the reference period (2008–2027). We perform Welch's t-test on the ensemble mean of the annual mean temperature, precipitation, and P-E during the year 2050–2069 to evaluate whether these regional changes are statistically significant. Since this test assumes that sampled data are independent, we perform the t-tests using the estimated effective sample size by assuming temperatures, precipitations, and P-E all follow a first-order autoregressive (AR(1)) process (Wilks, 2019), and perform multiple testing correction on the t-test results to account for spatial correlation using the FDR

method. We also evaluate how well these strategies compensate for the regional changes under climate change by comparing the area-weighted root mean square (rms) change.

Figure 8 shows the simulated changes in surface air temperatures. In SSP2-4.5, most areas on the Earth are warmer than the reference period, with the largest warming found in the Arctic region due to Arctic amplification. Overall, the temperature

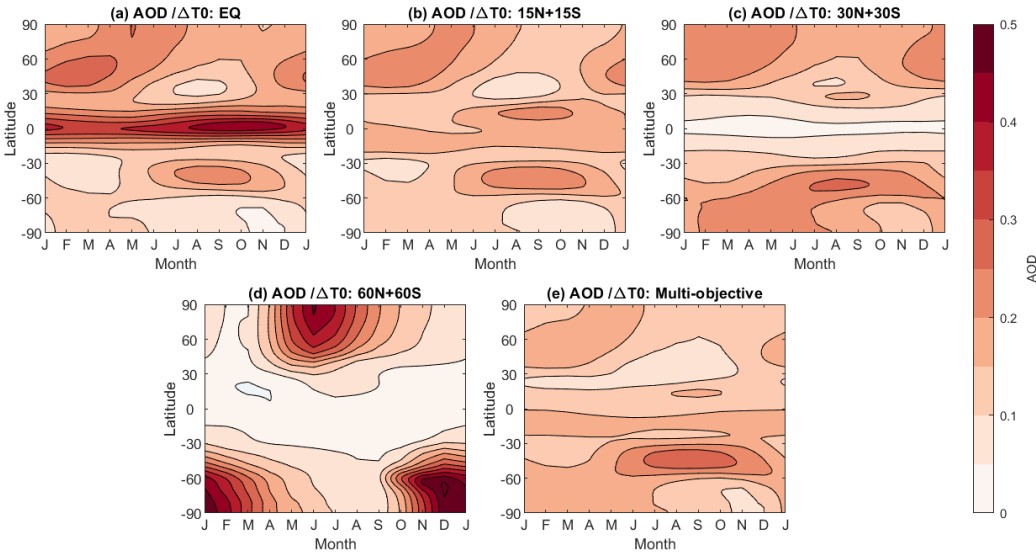

**Figure 7.** Simulated seasonal cycle of AOD at each latitude for $1^\circ$C of global mean cooling under each SAI injection strategy.

increase over land is higher than over the ocean (Fig. 8a). The exception to the overall warming trend is a region in the North

Atlantic Ocean which shows a cooling pattern (so called 'North Atlantic warming hole') that is related to the weakening of Atlantic meridional overturning circulation (AMOC) (see Tilmes et al. (2020) and Fasullo and Richter (2023); see also Fig. 16). The North Atlantic warming hole has also shown up in simulations in other climate models (Chemke et al., 2020; Keil et al., 2020) as well as in the RCP8.5 scenario simulated in CESM1(WACCM) (Tilmes et al., 2017). In addition to the reduced northward heat transport due to the weakening of AMOC, the formation of the warming hole has been shown to be also driven

by increased ocean heat transport from the warming hole to higher latitudes and a shortwave cloud feedback (Keil et al., 2020).

Figures 8(b)-(f) show that all SAI strategies effectively counteract the large-scale surface warming, as illustrated by the large fraction of surface area showing no statistically significant temperature difference relative to the reference climate. With SAI, the percentage of area with no statistically significant change ranges from 71 % to 84 %, while only 15 % of total area has no statistically significant difference without SAI. Despite similar magnitudes of global mean cooling (Fig. 1(a)), different SAI

strategies yield different regional temperature responses. The EQ strategy undercools the Southern Hemisphere, which is due to greater radiative heating that needs to be mitigated in SH in CESM2(WACCM6) (Fasullo and Richter, 2023). In contrast, the 60N+60S strategy overcools the Arctic and undercools the tropics because the injections are focused at higher latitudes and the resulting aerosols are rapidly transported poleward and downward by the Brewer-Dobson circulation.

In all simulations (Fig. 8), the surface air temperature in a region in the North Atlantic Ocean is lower than the reference

period, similar to the response found in the SSP2-4.5 simulation. This phenomenon is caused by the weakening of AMOC, which is discussed above and in more detail in Section 4.8. We also find consistent temperature changes over the Pacific Ocean across all SAI simulations and the SSP2-4.5 simulation, with relative warming in the eastern Pacific in both its equatorial

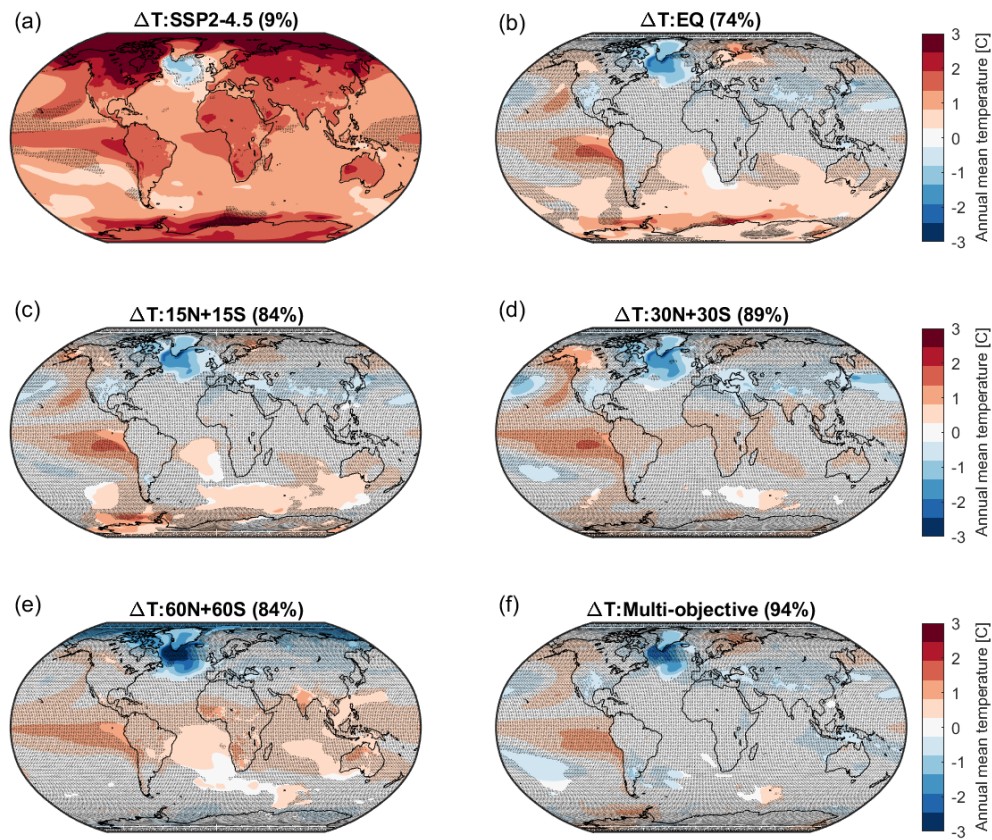

**Figure 8.** Changes in surface air temperature, averaged over 2050–2069, compared to the reference period (2008–2027) for (a) SSP2-4.5 and (b)-(f) different SAI injection strategies. Shaded areas indicate where the change relative to the reference period is not statistically significant based on a two-tailed Welch's t-test with a confidence level of 95 %. The percentage of area with no statistically significant change in surface air temperature is listed in the title of each map.

and northern regions compared to the reference period, albeit differing in the strength and horizontal extent of the anomalous equatorial Pacific warming. The pattern is similar to the pattern associated with the positive phase of the El-Nino Southern Oscillation (ENSO; e.g., McGregor et al., 2022), and as such projects on changes in the ENSO index. This is associated with changes in the strength and the position of the Walker Circulation (Bednarz et al., 2023a), contributing to the precipitation changes simulated in the Amazon region (Section 4.4). As pointed out in Visioni et al. (2023b), such changes are also dependent on the choices of reference period against which we are comparing, and are in part driven by an under-compensation of GHG warming.

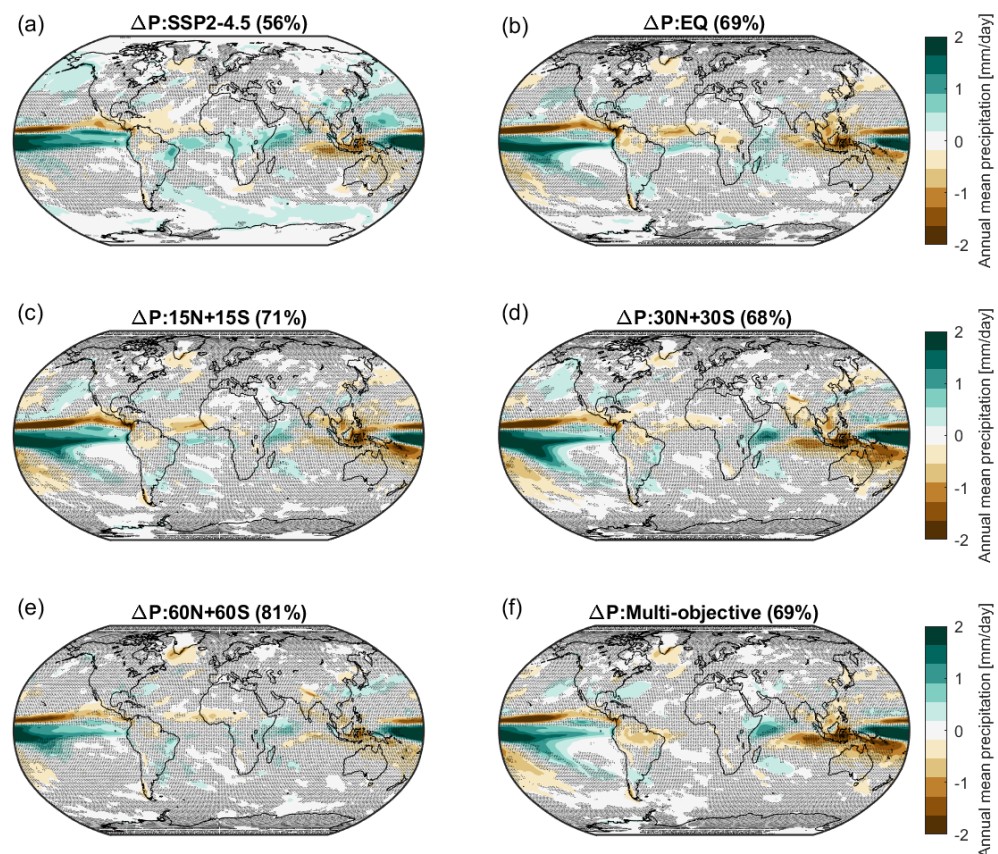

**Figure 9.** Changes in precipitation averaged over 2050–2069, compared to the reference period (2008–2027) for (a) SSP2-4.5 and (b)-(f) different SAI injection strategies. Shaded areas indicate where the response is not statistically significant based on a two-tailed Welch's t-test with a confidence level of 95 %. The percentage of area with no statistically significant change in precipitation is listed in the title of each map.

Figure 9 shows the simulated changes in precipitation. Under the SSP2-4.5 scenario, about 43 % of the area has a statistically significant change in precipitation compared to the reference period (Fig. 9a). While the percentage of area with statistically significant change in precipitation (27–38 %) is slightly reduced by SAI, SSP2-4.5 and SAI scenarios share similar spatial patterns of changes in precipitation. In particular, among SSP2-4.5 and all SAI cases, the most significant change occurs in the equatorial Pacific Ocean and follows a similar pattern – i.e., precipitation decreases in the northern region and increases in the southern region. This corresponds to the ITCZ shifts discussed in Section 4.5, and the fact that none of the SAI strategies manage to fully offset the southward ITCZ shift simulated in SSP2-4.5.

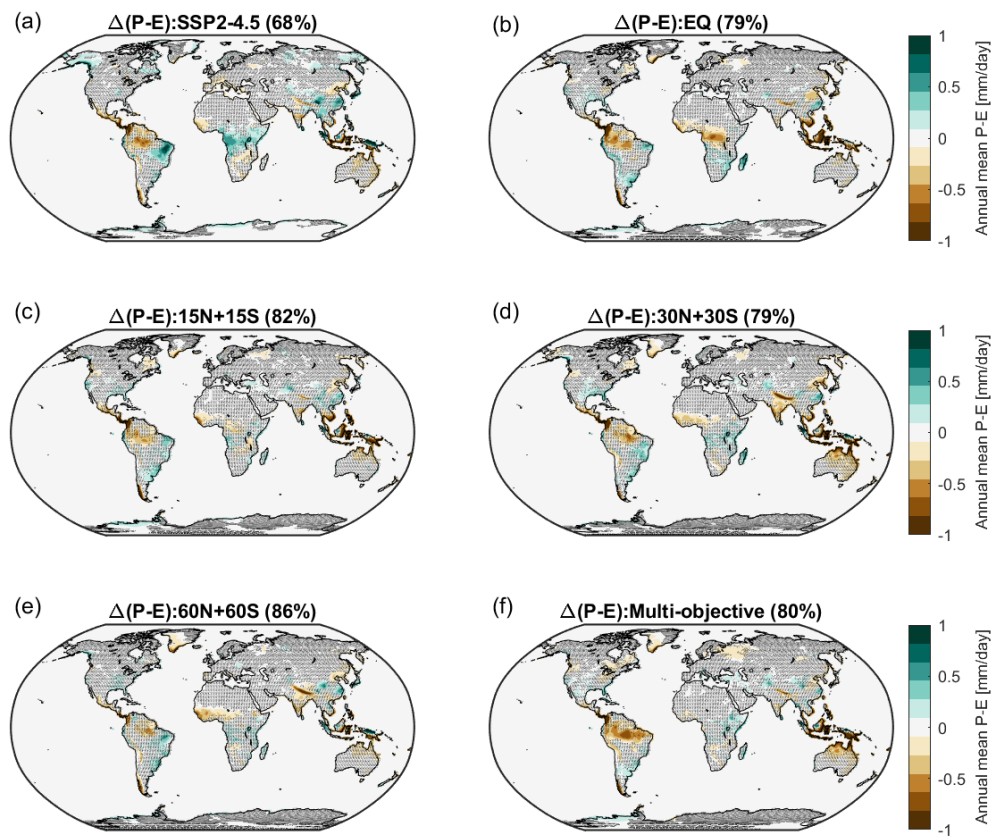

**Figure 10.** Changes in precipitation minus evaporation (P-E) over land averaged over 2050–2069, compared to the reference period (2008–2027) for (a) SSP2-4.5 and (b)-(f) different SAI injection strategies. Shaded areas indicate where the response is not statistically significant based on a two-tailed Welch's t-test with a confidence level of 95 %. The percentage of land area with no statistically significant change in P-E is listed in the title of each map.

The net flux of water from the atmosphere to the Earth's surface is described by precipitation minus evaporation (P-E). Figure 10 shows the simulated changes in P-E over land. Under the SSP2-4.5 scenario, 33 % of the land area has a statistically significant change in P-E compared to the reference period (Fig. 10a) and the percentage of land area with statistically significant change in P-E is slightly reduced by SAI (Fig. 10(b)-(f)). While the SAI scenarios have roughly the same percentage of the land area with statistically significant change in P-E (20–27 %), the regional changes in P-E vary between the different SAI strategies as well as the SSP2-4.5 run (Fig. 10). For example, the EQ strategy makes central Africa drier, while the P-E response in central Africa is not statistically significant under other SAI strategies. Also, the reduction in P-E over North India is statistically significant under 30N+30S and 60N+60S strategies, but not statistically significant under the other strategies.


To evaluate how well these strategies compensate for the change in regional temperature, precipitation, and P-E over land under climate change, we also calculate an ensemble mean area-weighted rms change comparing the 2050–2069 average to the reference period (Fig. 11(a)). We also calculate the rms change due to natural variability alone. This is done by first detrending the annual mean over 2008–2027 at each gridbox in the three ensemble members, and then calculating the area-weighted rms standard error of the processed data assuming an AR(1) autocorrelation process. If an SAI strategy fully compensates the

GHG-induced regional changes, then on average the rms response will be similar to the rms change due to natural variability alone. However, we find that in all SAI strategies, the rms temperature change is larger than the rms temperature change that one would expect due to natural variability alone (i.e., $0.15°\,$C, represented by the dashed line in Fig. 11(a)), indicating imperfect compensation of the pattern of warming under climate change. Among the SAI strategies considered here, the multi-objective strategy best minimizes the spatial rms of temperature changes, as indicated by the lowest rms temperature change

(rms T=$0.38°\,$C). The 60N+60S strategy results in an uneven cooling with the highest rms temperature change (rms T=$0.57°\,$C), but still much smaller than the rms temperature change in SSP2-4.5 without SAI (rms T=$1.53°\,$C). All SAI strategies give rise to mean rms precipitation and P-E responses that are larger than those from natural variability alone (which are estimated as approximately $0.16\ \mathrm{mm\ day}^{-1}$ and $0.10\ \mathrm{mm\ day}^{-1}$ for precipitation and P-E changes, respectively (Fig. 11(b)-(c)).The difference in the spatial rms of precipitation and P-E responses between SAI simulations and the SSP 2-4.5 simulation is notably

smaller than the difference in rms temperature responses, indicating poorer compensation of these metrics than temperature. When comparing any SAI strategy with the SSP2-4.5 case, the difference in rms precipitation response is no more than 30 % (Fig. 11(b)), and the difference in rms P-E response over land is no more than 12 % (Fig. 11(c)).

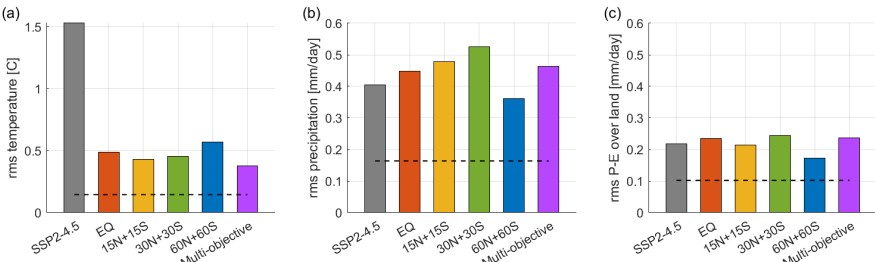

**Figure 11.** Area-weighted root mean square deviation between the (a) temperature, (b) precipitation, and (c) P-E over land averaged over 2050–2069 and the reference period (2008–2027). The dashed lines represent the area-weighted root mean square of each quantity due to natural variability alone.

.

## 4.4   Precipitation change in Amazon and Congo Basins

In this section, we focus on the Amazon and Congo Basins in particular, to show that different strategies have different impacts

on regional precipitation. For the Amazon Basin, we average precipitation over the region between $5°\,$N - $15°\,$S and $50°\,$W - $78°\,$W (a total land area of $7.2 \times 10^6\ \mathrm{km}^2$). For the Congo Basin, we average temperature over the region between $8°\,$N - $10°\,$S

and $12°$ E - $31°$ E (a total land area of $4.6 \times 10^6$ km$^2$). Precipitation changes in these tropical river basins have direct effects on local ecosystems. Rainforests in both regions act as carbon sinks and are thus of great importance to global climate. It is well studied that El Niño Southern Oscillation (ENSO) is one of the main drivers of interannual variability in convective precipitation over the Amazon Basin (Marengo and Espinoza, 2016; Jiménez-Muñoz et al., 2016). Precipitation over the Amazon Basin is suppressed during El-Niño events and enhanced during La Niña events (Marengo and Espinoza, 2016; Jiménez-Muñoz et al., 2016).

In the Amazon Basin, the 20-year average (2050-2069) under SSP2-4.5 is similar to the reference level (Fig. 12(a)), though with regional variations within the basin; the central region becomes drier while the southeast area gets wetter (see precipitation maps in Fig.S5 in the supplementary material). All SAI strategies result in a reduction in the mean precipitation, except for the 60N+60S case (which is not statistically significantly different from either the reference or the SSP2-4.5 case). The multi-objective strategy yields the strongest precipitation reduction. The hemispherically-symmetric strategies show a dependence of the precipitation reduction on the latitude of injection, with the largest decrease in the Amazon Basin precipitation in EQ and no statistically significant decrease in 60N+60S. This pattern of precipitation changes is likely related to the corresponding changes in the intensity of the tropospheric Walker Circulation and, thus, ENSO response, as also discussed in (Bednarz et al., 2023a). We approximate the ENSO changes by calculating the ENSO index as a difference in near-surface air temperature between the Nino 3.4 region (5N-5S, 120W-170W) and all tropical oceans (20N-20S), based on the method described in van Oldenborgh et al. (2021). The strength of the Walker Circulation is approximated by the difference in sea-level pressure between the East Pacific Ocean (5N-5S, 80-160W) and the Indian Ocean (5N-5S, 80-160E), based on the method described in Kang et al. (2020). Figure S7 in the supplementary material shows that changes in the Nino 3.4 index and the strength of Walker Circulation both contribute to and partly explain the precipitation responses simulated across the different SAI strategies and the SSP2-4.5 simulation in the Amazon Basin, with the coefficient of determination ($R^2$) of the best-fit linear regression functions equal to 0.62 and 0.66, respectively.

In the Congo Basin, the average precipitation in the SSP2-4.5 scenario increases over time (Fig. 12(b)), likely as the result of the intensification of the global hydrological cycle under increasing surface temperatures (Section 4.1). In contrast, all SAI strategies result in a reduction in the mean precipitation in the Congo Basin compared to the SSP2-4.5 case as the global mean surface temperatures are reduced to around the reference level (Fig. 1(a)). While the multi-objective strategy brings the 20-year average (2050-2069) mean precipitation back to the reference level, other strategies either undercompensate or overcompensate the precipitation. The equatorial and 15N+15S injection strategies result in statistically significant undercompensation of the Congo Basin precipitation compared to the reference period, while 30N+30S and 60N+60S result in a small overcompensation. The dependence of the precipitation reduction in the Congo Basin on the latitude of aerosol injection is partly indicative of the corresponding impacts from the intensity change of the tropospheric Hadley Circulation. As shown in Bednarz et al. (2023a), Hadley Circulation weakens significantly under EQ and 15N+15S strategies but stays unchanged for 30N+30S and 60N+60S; these tropospheric circulation changes could thus contribute to and partially explain the precipitation changes simulated in the Congo Basin across the strategies.

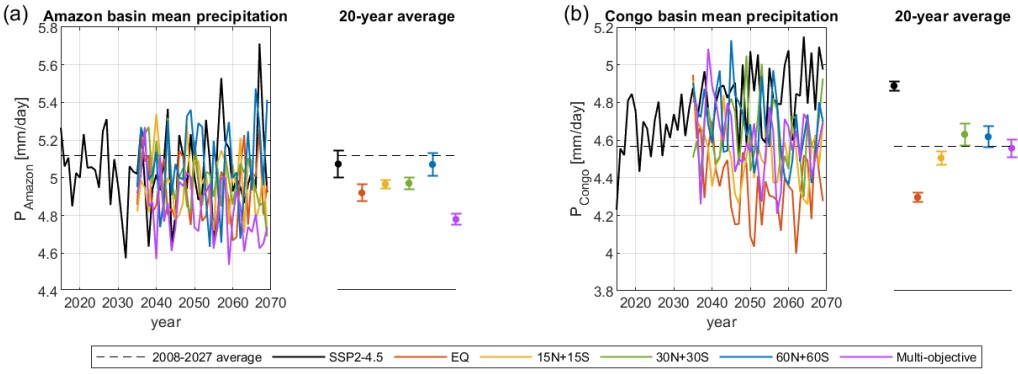

**Figure 12.** Time evolution of mean precipitation in (a) Amazon Basin and (b) Congo Basin. Each solid line represents the ensemble mean of each injection strategy. The dashed line represents the 20-year average during the reference period (2008–2027). The dots on the right of each panel represent the 20-year average over 2050–2069; the uncertainties in the calculated 20-year averages are estimated by $\pm 1$ standard error, and represented by the error bars.

## 4.5 Intertropical convergence zone

The Intertropical Convergence Zone (ITCZ) is a region of heavy precipitation near the equator, where the northeast and southeast trade winds collide (Byrne et al., 2018). Different metrics have been used in previous studies to define the ITCZ location, such those based on the precipitation centroid (e.g., Frierson and Hwang, 2012; Donohoe et al., 2013; Byrne et al., 2018; Lee et al., 2020) or based on atmospheric mass circulation (e.g., Hari et al., 2020; Cheng et al., 2022). Here, we define the ITCZ location as the latitude near the equator where the zonal mean meridional streamfunction at 500 hPa changes sign. The streamfunction at each latitude is calculated using the following equation:

$$\Psi = \frac{2\pi a cos(\phi)}{g} \int\limits_0^p [v]\, \mathrm{d}p' \tag{4}$$

where $[v]$ is the zonal mean meridional velocity, $a$ is the Earth's radius, $\phi$ is latitude, and $p$ is 500 hPa. The ITCZ location is approximated using linear interpolation of the centers of two consecutive grid cells that have meridional circulations of opposite directions.

Under GHG forcing alone, the latitude of ITCZ shifts southward from its location in the reference period (Fig. 13(a)). All hemispherically-symmetric SAI injection strategies shift the latitude of ITCZ further south, consistent with the stronger associated cooling in the NH than in the SH (Fig. 6(b)-(c)). The difference in the shift of ITCZ between the hemispherically-symmetric injection cases is modest, generally within one standard error. The multi-objective strategy, on the other hand, shifts the latitude of ITCZ northward from that due to GHGs alone, but still south of the ITCZ position in the reference period. The multi-objective strategy is the only one that explicitly targets hemispheric asymmetry; while $T_1$ is an imperfect proxy for managing ITCZ, it does result in improved compensation relative to the hemispherically-symmetric strategies, indicating the value of including an objective associated with asymmetric compensation.

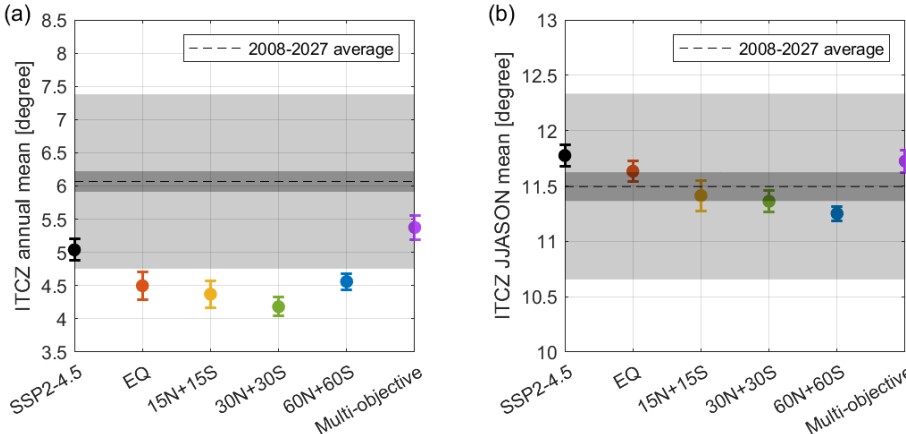

**Figure 13.** The 20-year (2050–2069) average (dots) and standard error (vertical bars) of the (a) annual mean and (b) seasonal mean (June through November, JJASON) latitude of ITCZ for SSP2-4.5 and the different SAI strategies. The dashed horizontal line represents the mean latitude of ITCZ during the reference period (2008–2027) and the shaded areas represent the corresponding standard error (dark gray) and standard deviation (light gray).

.

## 4.6 Tropical cyclone frequency

Existing studies show that climate change will decrease the overall tropical cyclone (TC) frequency but increase the frequency of the most intense ones (Bengtsson et al., 2007; Knutson et al., 2010; Camargo, 2013). Figure 14 evaluates the North Atlantic TC activity based on three TC indices that are described in Dunstone et al. (2013) and Jones et al. (2017). These TC indices evaluate the average precipitation in the main development region (MDR, defined as 5-20° N and 15-85° W), the inverse vertical zonal-wind shear between 850 and 250 hPa in the MDR, and the sea surface temperature (SST) difference between the MDR and the tropics as a whole. All three indices are calculated for the hurricane season in the North Atlantic, which is June–November (JJASON). An increase in MDR precipitation, inverse vertical zonal-wind shear, or the relative SST indicates an increase in TC frequency.

We find that all three TC indices show reduction in TC frequency under SSP2-4.5 (Fig. 14), in agreement with the existing literature (Bengtsson et al., 2007; Knutson et al., 2010; Camargo, 2013). TC frequency also decreases with SAI deployment, but the magnitude of reduction in TC frequency under different SAI strategies varies among the different TC metrics. In general, lower-latitude injections tend to have a larger reduction in the average MDR precipitation (Fig. 9), which yields a larger reduction in TC frequency compared to SSP2-4.5 or the higher-latitude injections (Fig. 14(a)). However, Fig. 14(b) shows that lower-latitude injections result in less increase in the zonal wind shear, which yields a smaller reduction in TC frequency compared to higher-latitude injections. The relative change in the inverse zonal wind shear between different SAI strategies is generally consistent with the relative change in ITCZ location in JJASON (Fig. 13(b)), as a southward shift of ITCZ is related

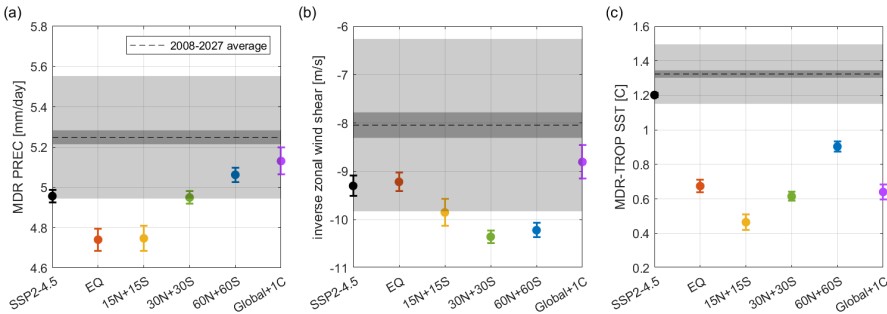

**Figure 14.** As in Fig. 10 but for the tropical cyclone frequency metrics: (a) the average precipitation in the main development region (MDR, see text for details), (b) inverse vertical zonal-wind shear in MDR, and (c) relative sea surface temperature difference between MDR and the tropics.

to an increase in zonal wind shear over the MDR (Dunstone et al., 2013). For the SST-based TC metric, we find that all SAI strategies result in substantially stronger reduction in TC frequency than those caused by climate change alone (Fig. 14(c)). The magnitude of the SST-based TC response in the geoengineering runs is smallest for the 60N+60S SAI strategy.

## 4.7 Arctic sea ice

The Arctic sea ice extent is expected to decrease in response to increasing global warming. If the current emissions of 40 Gt $yr^{-1}$ $CO_2$ continues without reduction, the Arctic Ocean is very likely to become ice free during summer before mid-century (Notz and Stroeve, 2018). The effectiveness of restoring Arctic sea ice through stratospheric aerosol injection is evaluated through comparing the predicted September Arctic sea ice extent (SSI) under SAI strategies and the SSP2-4.5 scenario. Figure 15(a) shows that all these five SAI strategies increase SSI to at least the reference period level by the year 2069. After around the year 2050, SSI starts to stabilize around the reference period level in the low- and mid-latitude injection cases, while SSI continues increasing in the high latitude injection case; the latter is consistent with the associated surface temperature changes (Fig. 8) and their equator-to-pole gradients (Fig. 1(c)). The 60N+60S strategy increases SSI by the highest amount; the 20-year (2050–2069) average of SSI is about $5 \times 10^6$ $km^2$, which is $1.4 \times 10^6$ $km^2$ more than the reference period level. The overcompensation of SSI in the 60N+60S strategy is mainly because of the largest fraction of aerosols found in the polar region.

## 4.8 Atlantic meridional overturning circulation

Section 4.3 and Fig. 8 show that all simulations yield a region in the North Atlantic Ocean that is cooler than the reference period. In accord, Fig. 16(a) shows that in CESM2(WACCM6), AMOC continues to weaken over the 21[st] century under SSP2-4.5, which is consistent with the predictions from other climate models (Chemke et al., 2020; Keil et al., 2020; IPCC, 2021). AMOC moves warm water northward at the surface from the tropics and cold water southward at the bottom of the ocean from

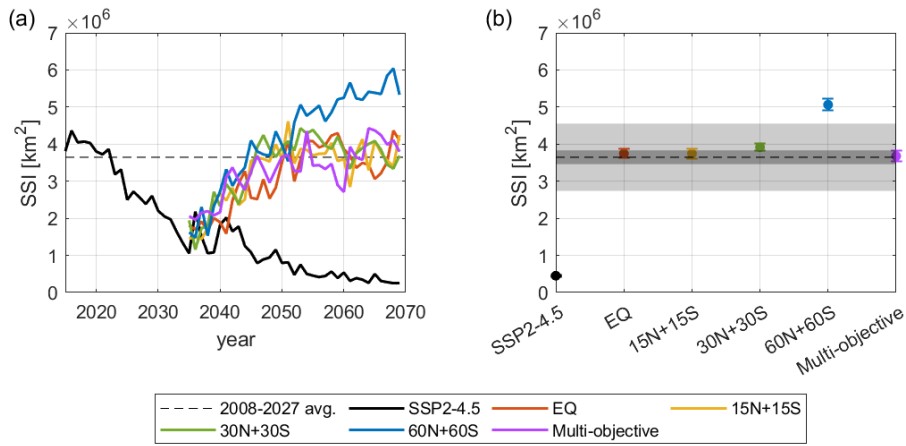

**Figure 15.** (a) Time evolution of September Arctic sea ice extent (SSI) for SSP2-4.5 and the different SAI strategies. (b) The 20-year (2050–2069) average (dots) and standard error (vertical bars) of SSI for SSP2-4.5 and the different SAI strategies. The dashed horizontal line represents the average SSI during the reference period (2008–2027) and the shaded areas represent the corresponding standard error (dark gray) and standard deviation (light gray).

.

the North Atlantic (Rahmstorf, 2002). As AMOC weakens, less heat is transported northward to the North Atlantic, which causes the decrease in the surface air temperature over that region (Danabasoglu, 2008).

We find that low- and mid-latitude injections are better at recovering AMOC than the high-latitude injections. Whilst the low- and mid-latitude injections do not restore AMOC back to the reference period, they do prevent further weakening of AMOC and keep AMOC at a strength similar to that in the year 2035 when injections are started. In comparison, AMOC continues weakening under the high-latitude SAI strategy, but at a much lower rate compared to the SSP2-4.5 case. The weakening of AMOC relative to the reference period is likely the main cause of the consistent cooling pattern over the North Atlantic in every strategy in Fig. 8.

## 5  Summary

The question of whether to deploy SAI requires not just one simple answer but a series of deliberate decisions, including decisions on how much cooling to provide, what other climate objectives to achieve, and how to achieve them. Understanding the differences in surface climate responses between different injection strategies is crucial for making informed decisions.

In this work, we have considered a set of five SAI strategies under the same climate and SAI scenario to explore the range of possible climate responses in one climate model. These include four hemispherically-symmetric injection strategies designed to maintain global mean temperature and one multi-objective strategy designed to maintain not only the global mean temperature but also the large-scale horizontal temperature gradients. The four hemispherically-symmetric strategies are $SO_2$ injection at

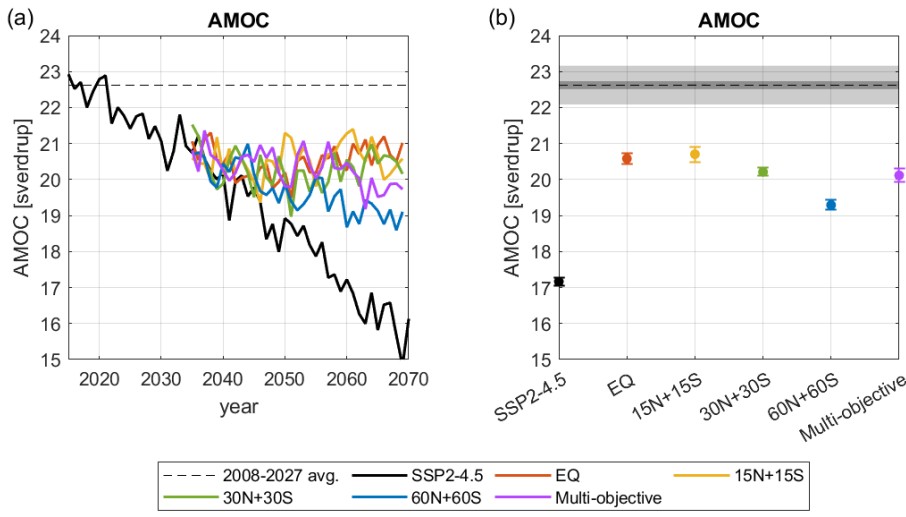

**Figure 16.** (a) Time evolution of the strength of the Atlantic Meridional Overturning Circulation (AMOC) under different SAI injection strategies over the period of 2035-2069, calculated as the maximum over depth and latitude of the meridional streamfunction in the North Atlantic. (b) As in Fig. 11(b) but for the strength of the AMOC.

.

the equator, and injections of equal $SO_2$ amounts at 15° N and 15° S, at 30° N and 30° S, and at 60° N and 60° S, the latter only during spring in each hemisphere.

475  The choice of SAI strategies notably affects the spatiotemporal distribution of aerosol optical depths (AOD) and injection efficiencies, and ultimately various surface climate responses. Injecting $SO_2$ in the mid-latitudes provides more cooling per unit of injection than injecting in either the tropics or high latitudes. The low efficiency in the equatorial injection is primarily due to larger sizes of aerosols formed. The low efficiency in the high-latitude injection case is due to the aerosols having a much shorter lifetime. On the other hand, the 60N+60S case yields the highest global cooling per unit of global mean AOD.

480  We find that while all of these five SAI strategies maintain the global mean temperature at the reference level, they also overcompensate the interhemispheric temperature gradient. The amount of reduction in the equator-to-pole temperature gradient depends on the choice of SAI strategy, with the high latitude strategy yielding most reduction. In addition, all strategies overcompensate global mean precipitation except the 60N+60S case. This is because injecting at lower latitudes results in stronger tropical cooling and more stratospheric heating, both of which lead to more reduction in precipitation.

485  Compared to the SSP2-4.5 case, all SAI strategies effectively reduce the percentage of area with statistically significant changes in temperature relative to the quasi present-day reference period, as well as the area-weighted root mean square (rms) change in regional temperature. In contrast, SAI strategies do not consistently reduce the rms change in precipitation minus evaporation (P-E) over land, nor the rms precipitation changes; the 15N+15S and 60N+60S strategies decrease the rms P-E change over land, while the other strategies slightly increase it.

The results show that while all SAI simulations reduce the weakening of the Atlantic meridional overturning circulation that is otherwise found for SSP2-4.5, they also fail to restore it back to the reference period level. Regarding September Arctic sea ice (SSI), all SAI strategies restore SSI back to the reference period level, except the high-latitude injection strategy, which overcompensates SSI. The responses in the location of intertropical convergence zone and tropical cyclone frequencies vary among different SAI strategies.

## 6   Discussion

Assessing the possible outcomes of SAI requires a good understanding of the possible impact from both the scenario and the choice of injection strategy. MacMartin et al. (2022) and Visioni et al. (2023b) have explored how different scenarios affect the climate responses to the same SAI strategy. In this work, we have demonstrated that different SAI strategies with similar objectives and under the same scenario would also affect the surface climate differently, with different distributions of outcomes. The study of these two different dimensions in the SAI design space lays the foundation for understanding the fundamental limits of SAI. Future research will explore combinations of these strategies, along with additional single-latitude cases (Visioni et al., 2023a; Lee et al., 2023a), to identify an optimal strategy for a given set of climate goals, and assess the underlying trade-offs between different climate goals, as well as to conduct similar analyses in other climate models. Knowing the range of possible climate outcomes and the trade-offs will help make informed decisions on future policy on SAI deployment. Ultimately, other factors besides climate outcomes are also needed to be considered when evaluating benefits and risks of SAI.

In addition, our study demonstrates that the multi-objective strategy (Kravitz et al., 2017; Tilmes et al., 2018; Richter et al., 2022) yields smaller residual regional temperature response than the hemispherically-symmetric strategies considered here. However, such a strategy requires adjusting injection rates across four different latitudes to manage multiple goals, and can thus be challenging to implement across many climate models. Simpler hemispherically-symmetric strategies would be easier to replicate in a large multi-model intercomparison: either the combined 15N+15S or 30N+30S case considered here may represent a reasonable trade-off between how well a strategy compensates for climate changes, and complexity of implementation in a climate model. Our study thus provides fundamental understanding of the differences in the resulting climate responses between the more complex multi-objective strategy and simpler hemispherically-symmetric ones, and as such is directly important for designing and understanding future large inter-model intercomparisons, including the next (seventh) phase of the Geoengineering Model Intercomparison Project (GeoMIP).

It is important to note that all simulations considered here are conducted using a single climate model, namely CESM2(WACCM6). Different climate models yield different patterns of AOD and surface climate responses for the same injection strategy (Visioni et al., 2023a; Fasullo and Richter, 2023). Also, atmospheric and climate responses from strategies with different injection locations are subject to different model structural uncertainties (e.g., Visioni et al., 2023a; Bednarz et al., 2023b). Simulating the same set of injection strategies in different global climate models will thus be important for better characterizing the uncertainties. In addition, the current study uses only a limited number of climate metrics to compare the different SAI strategies; other

aspects of climate that are not analyzed here (e.g., Antarctic ice sheets, permafrost carbon, sea level, and ozone), may provide additional insights on the benefits and risks of SAI.

*Data availability.* Data for the new simulations presented in this study are available at https://doi.org/10.5281/zenodo.7545452 (Zhang et al., 2023). Data for multi-objective strategy (from Visioni, 2022) are available at https://doi.org/10.7298/xr82-sv86.

*Author contributions.* YZ conducted all analyses and wrote the paper with editing from DM, EB, DV and BK; YZ and DM conceived the study with input from all authors and EB and DV assisted with conducting simulations.

*Competing interests.* The authors declare that they have no conflict of interest.

*Acknowledgements.* The authors would like to acknowledge high-performance computing support from Cheyenne (https://doi.org/10.5065/D6RX99HX) provided by NCAR's Computational and Information Systems Laboratory, sponsored by the National Science Foundation. Support for Y. Zhang and D. G. MacMartin was provided by the National Science Foundation through agreement CBET-2038246. Support for D. Visioni and E.M. Bednarz was provided by the Cornell Atkinson Center for a Sustainable Future. Support for BK was provided in part by the Indiana University Environmental Resilience Institute. The Pacific Northwest National Laboratory is operated for the U.S. Department of Energy
by Battelle Memorial Institute under contract DEAC05-76RL01830. The CESM project is supported primarily by the National Science Foundation.

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
