# Peer review of "Hemispherically-Symmetric Strategies for Stratospheric Aerosol Injection"

_EGUsphere, 2023_

## Referee Comment (RC1)

[referee-annotated manuscript omitted]

---

## Referee Comment (RC3)

**Comments on "Introducing a Comprehensive Set of Stratospheric Aerosol Injection Strategies" by Yan Zhang al.**

The manuscript presents an inter-comparison of different comprehensive set of stratospheric aerosol injection (SAI) strategies with the background emission scenario from the Shared Socioeconomic Pathway (SSP) 2-4.5 using WACCM climate model experiments. The manuscript evaluates the injection rates as well as the impact of SAI on near-surface air temperature, precipitation, Arctic sea ice, ITCZ, AMOC and tropical cyclone frequency. The information is very useful as the world is slowly acting to meet the Paris agreement on time to avoid severe climate impact and hazards. Although the manuscript contains some interesting material, which should be published, it could be significantly improved qualitatively in some parts (introduction and results). Some paragraphs and sections are poorly discussed, therefore, they need to be revised by enhancing the discussion about the scientific content, the structure of results presentations as well as combining certain figures to ease the understanding of the manuscript findings and to improve the quality of the manuscript. Particularly, the precipitation differences are overlooked. 30% changes of precipitation in keys regions such as Amazonia forest and Congo basin will significantly impact wildlife and flora in these region as well as the forest ability to absorb atmospheric $CO_2$ as  SAI has zero effect on $CO_2$ removal. The precipitation changes overland are much important to investigate because food security, agriculture and so many others vital component for human survival.

I recommend major revisions. In the following here are my major and specific points as well as general concerns:

**Major points:**

1. The surface climate response to different SAI strategies is present with not much caution know the role of the impact of model inter-annual variability on the distribution of SAI into the stratosphere as well as its feedback on surface climate. According to Bittner et al (2016), one need 7 ensembles in the tropics and 40 ensembles in the extra-tropics to capture accurately model circulation response to SAI, therefore, the related feedback to surface climate. There is a need to be caution on how to discuss the

findings here. More than 3 ensembles very like needed to constrained model internal variability.

2. The manuscript overlooks the impact of SAI strategies on precipitation and ITCZ, particularly in key region such as Amazonia and Congo Basin, which are key regions for human. Such as "the difference is no more than 30% (page 14, line 295)" are misleading regarding the interpretation of the SAI strategies on precipitation. Amazonia is responsible of 30% of oxygen production on Earth and is estimated to absorb some 2 billion tons of $CO_2$ per year, meaning that it soaks up about 5% of the world's total carbon emissions. The peat swamp forest of the Congo Basin stores around 29 billion tons of carbon, e.g. approximately equivalent to three years' worth of global GHG emissions, while the Basin as a whole absorbs nearly 1.5 billion tons of $CO_2$ a year. Therefore, I recommend to add two specific figures (like figure 1d) of precipitation changes under SAI strategies and SSP2-4.5 scenario for the Section 4.3.2. Precipitation

3. The discussion on the regional and global impact is mingled, therefore, I would like to suggest to restructure the results section 4 as following:
   a. 4. Results

      4.1 Large-scale g…..

      4.2  Injection rates and…

      4.3. Global surface climate response (fig 7 and fig 9)

      4.4. Regional surface climate response
   b. Please reorder the subsection as the following. After the "precipitation minus evaporation" section as well as "ITCZ", please discuss "tropical Cyclone frequency" and then followed by "SSI" and "AMOC".

4. Regrouping several figures is necessary here to ease the clarity and understanding the result better. Figure 8 and Figure 10 need to be put together.

5. Moving most of the figures in the appendix into the main manuscript is necessary for clarity.

**Minor points:**

1. Page 2, line 51-58, this "the differences in surface climate responses between some SAI strategies are much easier to detect than between others" needs to be rephrase each strategy may depend on how many ensemble used for taking into account model internal variability, which can

induce different injection rates based on model and SAI strategy. Please rephrase it.

2. Page 3, line 79, How can you affirm this "…the conclusion is expected to be reasonably robust and model independent…" knowing the model internal variability is not constrained by observations? Please rephrase it.

3. Page 8, line 184-185, Please replace the sentence by "Figure 2 shows the evolution of the total SO2 injection rate in each SAI strategy (a), and the 20-year (2050–2069) average injection rates (b)."

4. Page 8, line 190-192 I wonder the role of the BDC on the "This hemispheric asymmetry in the distribution of SO2 injections is likely due to the rapid cloud responses to elevated CO2 levels in CESM2(WACCM6), resulting in greater radiative heating that needs to be mitigated in the SH (Fasullo and Richter, 2023)."

5. Page 9, line 221-222, this is misleading "This asymmetry arises as the northern hemisphere has a stronger Brewer-Dobson circulation than the southern hemisphere". The inter-annual variability, which is much larger in NH than in SH, is what causes the asymmetry as the BDC is stronger in SH than NH.

6. Page 10, line 229, please add "the seasonality in" before "the Brewer-Dobson circulation" you add these citations.

7. Page 10, line 229, please remove "also".

8. Page 12, line 265, this "as solar reduction doesn't significantly change the Walker Circulation" is not clear. Please clarify or remove it.

9. Page 12, line 275-279, this paragraph is not clear. Please rephrase it.

10. The result about precipitation responses in section 4.3.2 are overlooked. Please better discuss these results.

11. Page 13, line 292-293, this "The difference in rms … temperature responses." is not correct for Amazonia & congo basin (Fig 9).

12. Page 14, line 295, this "the difference is no more than 30 %" is really misleading as the precipitation changes as well as their importance on mainland and certain key regions are not homogeneously distributed.

13. Page 14, line 308-311, A discussion about the link between TICZ changes with different SAI strategies is missing.

14. Figures 8 and 10 should be combined for clear discussion and reduction the numbers. For instance global and regional plots separately.

15. There is needs for separating global and region response better from page 11 to the end.

16. Page 17, line 326, it is not figure Fig 11a but Fig. 12a.
17.  Page 17, Paragraph 338-341 is speculative. Please rephrase it.
18. Please move most of the Appendix figures into the main text discussed.

Reference:
@article{Bittner-2016,
author = {Bittner, Matthias and Timmreck, Claudia and Schmidt, Hauke and Toohey, Matthew and Krüger, Kirstin},
title = {The impact of wave-mean flow interaction on the Northern Hemisphere polar vortex after tropical volcanic eruptions},
journal = {Journal of Geophysical Research: Atmospheres},
volume = {121},
number = {10},
pages = {5281-5297},
doi = {https://doi.org/10.1002/2015JD024603},
url = {https://agupubs.onlinelibrary.wiley.com/doi/abs/10.1002/2015JD024603},
year = {2016}
}

WMO Ozone assessment chap about SAI for an overview about all different techniques already performed:
https://csl.noaa.gov/assessments/ozone/2022/downloads/Chapter6_2022OzoneAssessment.pdf

---

## Author Comment (AC1)

**Author Responses to Reviewer #1's Comments**

*Original referee comments are in italics in black*

Author responses are in blue

1. *"The study is not comprehensive".*

    We agree that the word "comprehensive" is not quite the right choice of word, and may not accurately describe the novelty of this manuscript. We have replaced that with the word "novel". We have changed the title to "Novel Hemispherically-Symmetric Strategies for Stratospheric Aerosol Injection".

    This study systematically explores how the choice of SAI strategy affects climate responses, which is a key dimension of the range of possible climate responses to SAI. This study describes four hemispherically-symmetric injection strategies, including three strategies that are introduced for the first time and one equatorial injection strategy. Previous studies only look at up to two strategies at the same time. Zhang et al. (2022) have estimated that there are 6-8 injection strategies that produce detectably different surface climate responses, when providing 1C global cooling. The selection of these four injection strategies is based on the conclusion in Zhang et al. (2022), and is explained in Line 33-66, and Line 76-111. We have modified the paragraphs in Line 76-111 to better justify the selection of these strategies.

2. *"The abstract is poorly written. It does not explain what global warming scenario is used. It does not mention what climate models are used."*

    We describe the global warming scenario and climate model in Section 2 and 3. The global warming scenario is SSP2-4.5, and cooling scenario is one that starts in 2035 and ramps down to a target of 1.0C above preindustrial; these are described in Section 3 – Simulations; we do not think it is appropriate to go into this level of detail in the abstract as it is not directly relevant to the conclusions. The climate model used is CESM2(WACCM6), which is described in Section 2 – Climate Model; we have added this information to the abstract.

3. *"It jumps right into SAI while ignoring the fact that it does not exist, and is only a proposed scheme."*

    In the first paragraph in the introduction section, we make it very clear that stratospheric aerosol injection (SAI) could be an option in the future. We never said that it has been implemented.

4. *"It ignores the need to assess a wide range of potential benefits and risks before it is ever implemented."*

    There are many possible design strategies for SAI. The purpose of this paper is to analyze how the surface climates respond to different strategies differently by simulating multiple new strategies that have not been looked at in existing studies. Knowing the surface

climate responses to different strategies is one component, and we agree only one among multiple, that helps evaluate the benefits and risks of injection aerosol in the stratosphere, and the fundamental limits of this approach.

In line 33-40, we wrote, "To inform future decisions on SAI deployment, it is important to have a sufficient understanding of the range of possible climate responses under SAI; these would depend on both the scenario and strategy… Collectively, these two studies capture two key dimensions of the range of possible climate responses to SAI". In Section 6, we add "Climate response is only one factor to evaluate in supporting future climate decisions; other factors are also needed to be considered in evaluating benefits and risks of SAI " to explicitly note that there are many factors that need to be considered in evaluating benefits and risks of SAI. We have also modified the abstract and explicitly note this in Line 3 (also see our response to the first point in #10).

5. *"It does not say what is being injected. In fact the experiments are injecting gas and not aerosol."*

The term "Stratospheric Aerosol Injection" is a term used by the NASEM 2021 report, "Reflecting Sunlight: Recommendations for Solar Geoengineering Research and Research Governance", and we prefer to adhere to that nomenclature. In the abstract, we mentioned that in our simulations, SO2 is being injected in the stratosphere. In Line 30-32, we mentioned "such an approach would consist of injecting aerosols, or their precursors, in the lower stratosphere to reflect a small fraction of the incoming solar radiation back to space, as a result, lowering the global mean temperature". In Line 32, we have added "In this paper, we focus on SO2 injection".

6. *"The scientific questions being addressed by this paper are obscure. The paper says it wants to examine the response to certain sulfur dioxide emissions with respect to one global warming scenario using one climate model and specified injection altitudes and temperature reduction goal. It is by no means comprehensive. But why are they doing it?"*

We believe that it is important to explicitly show that injection strategy plays an important role when assessing the stratospheric aerosol injection approach, and to understand how different the climate responses would be under different injection strategies. This is relevant not just for understanding how the climate responds, but for ultimately being able to understand trade-offs and fundamental limitations, which is clearly more than any single paper can ever do.

We disagree with the reviewer that these '*scientific questions being addressed by this paper are obscure*'. In fact the crucial role of injection strategy in determining the simulated climate response to SAI has been recognized by many published studies in existing literature; those studies used either fixed-amount single-latitude injection simulations (Richter et al., 2017; Tilmes et al., 2018; Visioni et al., 2023; Bednarz et al., 2022; 2023) or a pair of equatorial vs multiobjective strategies (Kravitz et al., 2019; Visioni et al, 2021). While single-latitude injection strategies at mid-high latitudes are useful for understanding underlying physical processes, those are unlikely to represent responsible long-term deployment strategies due to the expected

large effect on ITCZ. While the second set of studies (Kravitz et al., 2019; Visioni et al., 2021) demonstrate that the importance of injection location also holds for more complex strategies, it is paramount to explore a broader range of such injection strategies.

As noted earlier we agree that "comprehensive" was not the correct word choice. Nonetheless, as we expect (in this model, and at least plausibly in others) that there are of order 6-8 distinct choices, the linear combination of a relatively small number of distinct strategies would span the range of possible climate responses. The selection of these four injection strategies is based on the conclusion in Zhang et al. (2022), and is explained in Line 33-66, and Line 76-111. This set of strategies would include the 4 new strategies simulated and evaluated here, the multi-objective one also evaluated here, and some more asymmetric strategies. We have rewritten the more general motivation in the introduction, and the specific motivation for these particular choices in Section 3.

7. *"The paper is very long and detailed, going through many variables from the climate model simulations they did. I lost interest before I got halfway through."*

We believe it is important to assess how the strategy affects a range of climate outcomes. Different strategies affect different climate variables differently. For example, a strategy may overcompensate some climate variables but undercompensate others. It is important to not only look at one or two climate variables, but a larger set of climate variables, and evaluate how these strategies studied here affect each climate variable.

8. *On line 65 the paper says, "The understanding that comes from the analysis of the differences between these strategies lays the foundation for future work." That is what a technical report should be doing, not a journal article which needs new science to justify publication.*
   We do not agree with this comment. New research is always built upon the understanding and scientific knowledge from existing research; any good research paper should both have new science and lay a foundation for future work; the latter being true is a good thing. This study shows how the choice of SAI strategies impacts surface climate responses, which is necessary and novel knowledge that future research on evaluating the benefits and risks of SAI needs to be built upon.

9. *In several places, "We note that" is in the text and should be deleted. Every sentence should be noted or it should not be in the paper.*
   We deleted "We note that" in Line 87 and Line 108.

10. *There are 45 additional comments in the annotated manuscript. If the authors chose to reply to this review, a response of "we addressed all the comments" would not be sufficient. Each comment should be listed with a specific response.*
    Below are our responses to the additional comments in the annotated manuscript:
    1. Line 2: Change "Different injection strategies" to "Different stratospheric aerosol injection strategies". After "therefore, making informed future decisions on SAI requires an understanding of the range of possible climate outcomes", add ", in addition to other considerations".

2. Line 3: "therefore, making informed future decisions on SAI requires an understanding of the range of possible climate outcomes" is changed to "therefore, understanding the range of possible climate outcomes is crucial to making formed future decisions on SAI".
3. Line 6: The scope of this study is limited to climate goals. We have acknowledged that there are other possible goals in the discussion session.
4. Line 9: this is the name of the strategy.
5. Line 11: before "We…", we added "These strategies are simulated using the Earth system model CESM2(WACCM6-MA) with the global warming scenario SSP2-4.5"., in response to the comment on line 18.
6. Line 14: delete "notable".
7. Line 15: Replace "Among other findings" with "In addition".
8. Line 15: replace "in" with "in the".
9. Line 31: This sentence is meant to briefly explain how SAI works. As the reviewer suggested, there are many metrics for evaluating and quantifying the state of a climate. Because many metrics can be used to evaluate the effects from climate change and from SAI, we need to consider multiple metrics when evaluating the SAI strategies, which is exactly what we did in this paper.
10. Line 33: we thank the reviewer for valid points. However, other potential risks are beyond the scope of the current manuscript.
11. Line 39: "these two studies" are MacMartin et al. (2022) and this manuscript.
12. Line 41: change "can" to "could".
13. Line 42: use "would" in this sentence. "would overcool the tropical region and undercool…", and "would primarily cool…".
14. Line 51: change "do" to "potentially would".
15. Line 63: change "are" to "is".
16. Line 70: In line 71-73, we have described the chemistry in this model. In line 73, we have added the following sentence, "The ocean component is based on the Parallel Ocean Program Version 2 (POP2), the land component is Community Land Model Version 5 (CLM5), and the sea ice component is CICE5 (Danabasoglu et al., 2020)."
17. Line 79: We have modified the sentence as "Although the estimate of 6-8 distinct injection choices was made using CESM1(WACCM) simulations, the conclusion is expected to hold relatively well in CESM2(WACCM) due to similarities in the stratospheric circulation and aerosol microphysics between the two model versions. This is demonstrated by the results of a set of fixed-amount single-latitude injection simulations (Fig. S1 in supplementary material)". Using a completely different model is outside of the scope of this single-model study. However, we agree that cross-model validation should be done in future work.
18. Line 81: The word "pragmatic" here was intended to mean that we consider the strategies that are more likely to be considered for future deployment, in contrast to highly-asymmetric strategies that would notably change T1 and shift ITCZ. One can find many possible sets of 7 strategies that span the AOD design space of the same 7 degrees of freedom found in Zhang et al. (2022), which are single

latitude injections at 7 different latitudes: 60°N, 30°N, 15°N, the equator, 15°S, 30°S and 60°S. We chose this particular set of strategies based on not just the goal of spanning the same AOD design space, but also to focus only on strategies that would not result in large shifts in ITCZ. We agree that the word "pragmatic" was not a great choice of word as this word is somewhat subjective; different people may reach different conclusions regarding pragmatic. We have deleted this word in the manuscript.

19. Line 81: We thank the reviewer for the valid point. However, testing different altitudes or different particles is beyond the scope of this manuscript.

20. Line 94: both are grammatically correct.

21. Line 98: The sign of the hemispheric asymmetry in injection rates that would be needed to maintain T1 varies among different climate models. For example, in CESM1 more injection was required in the NH and in CESM2 more injection was required in the SH to compensate T1. We have rewritten this paragraph to better explain our motivation.

22. Line 102: The radiative forcing from $CO_2$ is roughly hemispherically symmetric, and thus to first order one might reasonably expect a hemispherically symmetric injection would compensate both T0 and T1. However, other effects - the fast cloud-adjustment to increased $CO_2$, as well as changes in AMOC and in tropospheric aerosol concentrations lead to changes in T1 that require asymmetric injection to compensate. These, however, are strongly model dependent (e.g., in CESM1 more injection was required in the NH and in CESM2 more injection was required in the SH to compensate T1). This model-dependency of the sign of the asymmetry in injection rates (NH-dominated or SH-dominated) is one reason we focus here only on symmetric strategies. Also see our responses in #21. We have rewritten this paragraph to better explain our motivation.

23. Line 104: These hemispherically-symmetric strategies that only target T0 are simpler to implement in many other climate models. It is relatively straightforward to tune one variable (overall injection rate) to meet one objective (T0) by hand. Simultaneously tuning multiple variables is more challenging without explicitly coding a feedback control algorithm. Simulations such as GeoMIP G6 demonstrate that modeling centers can adjust injection to meet one goal without needing to code a control algorithm. We have reworded the text to clarify.

24. Line 108: change "will" to "would".

25. Line 112: same response as in #1.

26. Table 1: We have added explanations for MAM and SON in Table 1 caption. T0, T1, and T2 are explained in line 91-92. We have changed "T0/T1/T2" to "T0, T1, T2".

27. Line 131: We have changed according to the reviewer's comment.

28. Line 142: We have changed according to the reviewer's comment.

29. Line 147: We have changed according to the reviewer's comment.

30. Line 159-163: We have removed our hypothesis on the overcompensation of T1.

31. Figure 1: We have increased the Font size in all figures.

32. Line 204: we have added citation: Butchart, 2014; Lee et al., 2021; Visioni et al., 2023.
33. Figure 6: We have added a Jan column after Dec.
34. Line 363-365: we thank the reviewer for valid points. However, the analysis of the above aspects is beyond the scope of the current manuscript. We have added "including" after "a series of deliberate decisions".
35. Line 366: We have changed the original sentence to "…a set of five SAI strategies…"
36. Line 396: same response as in #1.

---

## Author Comment (AC2)

**Author Responses to Reviewer #2's Comments**

*Original referee comments are in italics in black*

Author responses are in blue

*"Claiming that the investigation presented here is "comprehensive," in a way that no previous studies have been, is simply incorrect. The authors present 5 injection strategies. Four of these are actually the same strategy, but with different injection latitudes - so actually only 2 different strategies. This is very much in line with previous similar studies in the literature, including ones that share many of the same co-authors with this paper. So this choice of language is deeply puzzling."*

We agree with the reviewer that the word "comprehensive" is not quite the right choice of word, and have replaced that with the word "novel". We have changed the title to "Novel Hemispherically-Symmetric Strategies for Stratospheric Aerosol Injection".

This study systematically explores how the choice of SAI strategy affects climate responses, which is a key dimension of the range of possible climate responses to SAI. Our use of the term "strategy" is defined clearly on Line 36, and is the same as usage in other papers. Injecting at different latitudes and/or seasons are considered as different injection strategies; that is, the "strategy" describes all the different choices regarding how one meets a particular temperature target. This study describes four hemispherically-symmetric injection strategies, including three strategies that are introduced for the first time and one equatorial injection strategy. Previous studies only look at up to two strategies at the same time. Zhang et al. (2022) have estimated that there are 6-8 injection strategies that produce detectably different surface climate responses, when providing 1C global cooling. The selection of these four injection strategies is based on the conclusion in Zhang et al. (2022), and is explained in Line 33-66, and Line 76-111. We have modified the paragraphs in Line 76-111 to better justify the selection of these strategies.

*"Far too little information is given on the technical approach. This aspect of the paper reads like an internal report rather than a manuscript for the literature. The other reviewer also commented on this. More information is needed on the model and climate scenario underpinning the simulations."*

The climate model and global warming scenario are clearly explained in Section 2 and 3. The climate model used is CESM2(WACCM6), which is described in Section 2 – Climate Model. The global warming scenario is SSP2-4.5, which is described in Section 3 – Simulations. Additional details on the controller are added as described below.

*"The authors are surprisingly vague about the "controller(s)" which are used to determine injection rates. Equations and parameters for this technical feature need to be shared - along with some discussion of how this would be implemented in any kind of practical sense. The authors are directed to another paper for these details - which would not be sufficient even if the reference trail were clear - but it is not at all clear what paper is being referenced here (after 10 minutes searching I did not find a Visioni et al. 2022 with this title)."*

The current manuscript refers to the preprint of the Visioni et al. study that includes most of the important details behind the controller. Since this study has been accepted and published in 2023, the correct citation should be Visioni et al., 2023. We apologize for the confusion, and have now corrected the typo. Visioni et al. 2023 describes the sensitivity to injection, and MacMartin et al., 2014 and Kravitz et al., 2017 describe more generally how the controller is designed. We have added a paragraph describing the details of the controller implementation for the strategies described here.

---

## Author Comment (AC3)

**Author Responses to Reviewer #3's Comments**

*Original referee comments are in italics in black*

Author responses are in blue

*The manuscript presents an inter-comparison of different comprehensive set of stratospheric aerosol injection (SAI) strategies with the background emission scenario from the Shared Socioeconomic Pathway (SSP) 2-4.5 using WACCM climate model experiments. The manuscript evaluates the injection rates as well as the impact of SAI on near-surface air temperature, precipitation, Arctic sea ice, ITCZ, AMOC and tropical cyclone frequency. The information is very useful as the world is slowly acting to meet the Paris agreement on time to avoid severe climate impact and hazards. Although the manuscript contains some interesting material, which should be published, it could be significantly improved qualitatively in some parts (introduction and results). Some paragraphs and sections are poorly discussed, therefore, they need to be revised by enhancing the discussion about the scientific content, the structure of results presentations as well as combining certain figures to ease the understanding of the manuscript findings and to improve the quality of the manuscript. Particularly, the precipitation differences are overlooked. 30% changes of precipitation in keys regions such as Amazonia forest and Congo basin will significantly impact wildlife and flora in these region as well as the forest ability to absorb atmospheric CO2 as SAI has zero effect on CO2 removal. The precipitation changes overland are much important to investigate because food security, agriculture and so many others vital component for human survival.*

We thank the reviewer for their helpful comments; in addition to the specific responses below we will carefully go through the manuscript to clarify presentation; we have also added emphasis and figures in supplementary material regarding the precipitation changes in key regions.

*I recommend major revisions. In the following here are my major and specific points as well as general concerns:*

*Major points:*

*1.    The surface climate response to different SAI strategies is present with not much caution know the role of the impact of model inter-annual variability on the distribution of SAI into the stratosphere as well as its feedback on surface climate. According to Bittner et al (2016), one need 7 ensembles in the tropics and 40 ensembles in the extra-tropics to capture accurately model circulation response to SAI, therefore, the related feedback to surface climate. There is a need to be caution on how to discuss the findings here. More than 3 ensembles very like needed to constrained model internal variability.*

We appreciate the reviewer's concern as to the role of model internal variability in the inferred responses. However, we believe that we acknowledge and account for the uncertainty in the diagnosed responses. We examine a response to a continuous SAI forcing, with 20-years of data per ensemble member (so 60 years in total for a single SAI

strategy). The Bittner et al. study examined the vortex response to a Tambora eruption (so an instantaneous aerosols forcing) during a single year after the eruption; as such they required a much larger number of ensemble members to confidently diagnose the response.

While increasing the number of ensemble members will improve the estimate of the forced signal, when a difference between two strategies is large enough, we may not need more than one ensemble member in order to show that the difference in the responses between these two strategies is statistically significant and to explain the underlying mechanism. For example, the equator-to-pole temperature gradient (T2) in response to 60N+60S is notably different from T2 in response to other strategies; in this case, one ensemble member would be sufficient to show the difference in T2 between 60N+60S and other strategies. This notable difference is due to the offsetting of arctic amplification by providing more cooling at high latitudes.

We note the main purpose of our study is to introduce a set of novel SAI strategies and provide an overview of some of the main differences and similarities. As such, we chose to optimize the usage of computing time and simulate as many strategies as possible by running three ensemble members per each strategy, a compromise we believe is acceptable in this case.

*2.    The manuscript overlooks the impact of SAI strategies on precipitation and ITCZ, particularly in key region such as Amazonia and Congo Basin, which are key regions for human. Such as "the difference is no more than 30% (page 14, line 295)" are misleading regarding the interpretation of the SAI strategies on precipitation. Amazonia is responsible of 30% of oxygen production on Earth and is estimated to absorb some 2 billion tons of CO2 per year, meaning that it soaks up about 5% of the world's total carbon emissions. The peat swamp forest of the Congo Basin stores around 29 billion tons of carbon, e.g. approximately equivalent to three years' worth of global GHG emissions, while the Basin as a whole absorbs nearly 1.5 billion tons of CO2 a year. Therefore, I recommend to add two specific figures (like figure 1d) of precipitation changes under SAI strategies and SSP2-4.5 scenario for the Section 4.3.2. Precipitation*

We thank the reviewer for their comments. We have modified the sentence on Line 295 as "For the corresponding changes in precipitation over the whole Earth surface (i.e. both land and ocean), the difference in rms P-E response over land is no more than 30% when comparing any SAI strategy with the SSP2-4.5 case (Fig. 8(b)). Although the difference in these global metrics between two strategies might look small, differences in the regional changes could be quite important and need to be evaluated individually".

We have added precipitation plots for the Amazon Basin and Congo Basin and a discussion of the results into Section 4.4. We also add regional precipitation maps for Amazon Basin and Congo Basin in the Supplementary Material.

The following paragraphs are added to the section describing regional precipitation responses.

"In this section, we focus on the Amazon Basin and Congo Basin in particular, to show that different strategies have different impacts on regional precipitation. For the Amazon Basin, we average precipitation over the region between 5N - 15S and 50W - 78W (a total land area of 7.2x10^6 km^2). For the Congo Basin, we average temperature over the region between 8N - 10S and 12E - 31E (a total land area of 4.6 x10^6 km^2). Precipitation changes in these tropical river basins have direct effects on local ecosystems. Rainforests in both regions act as carbon sinks and are thus of great importance to global climate. It is well studied that El Niño Southern Oscillation (ENSO) is one of the main drivers of interannual variability in convective precipitation over the Amazon Basin (Marengo and Espinoza, 2016; Jiménez-Muñoz et al., 2016). Precipitation over the Amazon Basin is suppressed during El-Niño events and enhanced during La Niña events (Marengo and Espinoza, 2016; Jiménez-Muñoz et al., 2016).

In the Amazon Basin, the 20-year average (2050-2069) under SSP2-4.5 is similar to the reference level (Fig. 12(a)), though with regional variations within the basin; the central region becomes drier while the southeast area gets wetter (see precipitation map in the supplementary material). All SAI strategies result in a reduction in the mean precipitation, except for the 60N+60S case (which is not statistically significantly different from either the reference or the SSP2-4.5 case). The multi-objective strategy yields the strongest precipitation reduction. The hemispherically-symmetric strategies show a dependence of the precipitation reduction on the latitude of injection, with the largest decrease in the Amazon Basin precipitation in EQ and no statistically significant decrease in 60N+60S. This pattern of precipitation changes is likely related to the corresponding changes in the intensity of the tropospheric Walker Circulation and, thus, ENSO response, as also discussed in Bednarz et al. (2023). We approximate the ENSO changes by calculating the ENSO index as a difference in near-surface air temperature between the Nino 3.4 region (5N-5S, 120W-170W) and all tropical oceans (20N-20S), based on the method described in Oldenborgh et al. (2021). The strength of the Walker Circulation is approximated by the difference in sea-level pressure between the East Pacific Ocean (5N-5S, 80-160W) and the Indian Ocean (5N-5S, 80-160E), based on the method described in Kang et al. (2020). Figure S7 in the supplementary material shows that both changes in the Nino 3.4 index and the strength of Walker Circulation partly explain the change of precipitation in the Amazon Basin, with the coefficient of determination ($R^2$) of the best-fit linear regression functions equal to 0.62 and 0.66, respectively.

In the Congo Basin, the average precipitation in the SSP2-4.5 scenario increases over time (Fig.12(b)), likely as the result of the intensification of the global hydrological cycle under increasing surface temperatures (Section 4.1). In contrast, all SAI strategies result in a reduction in the mean precipitation in the Congo Basin compared to the SSP2-4.5 case as the global mean surface temperatures are reduced to around the reference level (Fig. 1(a)). While the multi-objective strategy brings the 20-year average (2050-2069) mean precipitation back to the reference level, other strategies either undercompensate or overcompensate the precipitation. The equatorial and 15N+15S injection strategies result in statistically significant undercompensation of the Congo Basin precipitation compared to the reference period, while 30N+30S and 60N+60S result in a small overcompensation. The

dependence of the precipitation reduction in the Congo Basin on the latitude of aerosol injection is partly indicative of the corresponding impacts from the intensity change of the tropospheric Hadley Circulation. As shown in Bednarz et al., 2023, Hadley Circulation weakens significantly under EQ and 15N+15S strategies but stays unchanged for 30N+30S and 60N+60S; these tropospheric circulation changes could thus contribute to and partially explain the precipitation changes simulated in the Congo Basin across the strategies."

List of references:

1.  Marengo, J. A. and Espinoza, J. C.: Extreme seasonal droughts and floods in Amazonia: causes, trends and impacts, International Journal of Climatology, 36, 1033–1050, https://doi.org/https://doi.org/10.1002/joc.4420, 2016.
2.  Jiménez-Muñoz, J. C., Mattar, C., Barichivich, J., Santamaría-Artigas, A., Takahashi, K., Malhi, Y., Sobrino, J. A., and Schrier, G. v. d.:Record-breaking warming and extreme drought in the Amazon rainforest during the course of El Niño 2015–2016, Scientific Reports, 6,33130, https://doi.org/10.1038/srep33130, 2016.

[Figure]

Fig 12. Time evolution of mean precipitation in (a) Amazon Basin and (b) Congo Basin. Each solid line represents the ensemble mean of each injection strategy. The dashed line represents the 20-year average during the reference period (2008--2027). The dots on the right of each panel represent the 20-year average over 2050--2069; the uncertainties in the calculated 20-year averages are estimated by ±1 standard error, and represented by the error bars.

[Figure]

Figure S5. Changes in precipitation (averaged over 2050–2069) in Amazon Basin compared to the reference period (2008–2027) for (a) SSP2-4.5 and (b)-(f) different SAI injection strategies. Shaded areas indicate where the response is not statistically significant based on a two-tailed Welch's t-test with a confidence level of 95%.

[Figure]

Figure S6. Changes in precipitation (averaged over 2050–2069) in Congo Basin compared to the reference period (2008–2027) for (a) SSP2-4.5 and (b)-(f) different SAI injection strategies. Shaded areas indicate where the response is not statistically significant based on a two-tailed Welch's t-test with a confidence level of 95%.

[Figure]

Fig S7. Correlation of change in precipitation in the Amazon Basin with (a) change in the Nino 3.4 index and (b) change in the strength of the Walker Circulation. Precipitation in the Amazon Basin is calculated as the average over the land region between 5N-15S and 50-78W. Nino 3.4 index is calculated as the difference in near-surface air temperature anomaly over the nino 3.4 region (5N-5S, 120-170W) and near-surface air temperature anomaly over all tropical oceans (20N-20S). The strength of Walker Circulation is calculated as the difference in sea-level pressure between the East Pacific Ocean (5N-5S, 80-160W), and the Indian Ocean (5N-5S, 80-160E). The change in these metrics is calculated as the difference between a 20-year average (2050-2069) and the reference period level (2008-2027). The error bars represent the standard error of the mean.

3. *The discussion on the regional and global impact is mingled, therefore, I would like to suggest to restructure the results section 4 as following:*

a. *4. Results*

*4.1 Large-scale g…..*

*4.2 Injection rates and…*

*4.3. Global surface climate response (fig 7 and fig 9)*

*4.4. Regional surface climate response*

b. *Please reorder the subsection as the following. After the "precipitation minus evaporation" section as well as "ITCZ", please discuss "tropical Cyclone frequency" and then followed by "SSI" and "AMOC".*

We thank the reviewer's suggestions on the results section, and have made the changes accordingly.

*4.    Regrouping several figures is necessary here to ease the clarity and understanding the result better. Figure 8 and Figure 10 need to be put together.*

We thank the reviewer for the comment. However, we don't think those two figures should be combined, as Figure 8 shows the rms changes of temperature and precipitation while Figure 10 shows ITCZ.

*5.    Moving most of the figures in the appendix into the main manuscript is necessary for clarity.*

We thank the reviewer for this comment. We have moved most figures in the appendix to the main paper, including combining what was Fig. A6 with Fig. 10 (on ITCZ). We have moved Figure A3-A5 to Supplementary materials.

*Minor points:*

*1.    Page 2, line 51-58, this "the differences in surface climate responses between some SAI strategies are much easier to detect than between others" needs to be rephrase each strategy may depend on how many ensemble used for taking into account model internal variability, which can induce different injection rates based on model and SAI strategy. Please rephrase it.*

We thank the reviewer for the suggestion: We have now rephrased the sentence to say: "The detectability of the differences in surface climate responses between SAI strategies depend on, among other factors, the level of global cooling and natural variability. While different SAI strategies do not result in the same surface climate, the differences in surface climate responses between some SAI strategies are much easier to detect than between others."

The reviewer is correct that natural variability will affect injection rates, yielding (slightly) different injection rates for each ensemble member.  However, the standard error of injection rates among three ensemble members for each SAI strategy is relatively small, so this is a small effect relative to the direct role of natural variability in assessing differences between strategies. Therefore, model internal variability does not affect the conclusion quoted here (line 51-52 in the manuscript). We have updated Figure 2(a) to reflect the standard error of the injection rates.

[Figure]

[Figure]

2. Page 3, line 79, How can you affirm this *"...the conclusion is expected to be reasonably robust and model independent..."* knowing the model internal variability is not constrained by observations? Please rephrase it.

We have modified the sentence as "Although the estimate of 6-8 distinct injection choices was made using CESM1(WACCM) simulations, the conclusion is expected to hold relatively well in CESM2(WACCM) due to similarities in the stratospheric circulation and aerosol microphysics between the two model versions. This is demonstrated by the results of a set of fixed-amount single-latitude injection simulations (Fig. S1 in supplementary material)". As the estimate is primarily determined by stratospheric circulation, it is reasonable to expect broadly similar number of distinct degrees of freedom in other models, but this needs to be validated.

3. Page 8, line 184-185, Please replace the sentence by *"Figure 2 shows the evolution of the total SO2 injection rate in each SAI strategy (a), and the 20-year (2050–2069) average injection rates (b)."*

We have modified the sentence as "Figure 2 shows the evolution of the total SO2 injection rate in each SAI strategy (Fig.2(a)), and the 20-year (2050–2069) average injection rates (Fig. 2(b))".

4. Page 8, line 190-192 I wonder the role of the BDC on the *"This hemispheric asymmetry in the distribution of SO2 injections is likely due to the rapid cloud responses to elevated CO2 levels in CESM2(WACCM6), resulting in greater radiative heating that needs to be mitigated in the SH (Fasullo and Richter, 2023)."*

The reviewer is correct that asymmetry in BDC does mean that even a hemispherically symmetric injection rate will lead to some asymmetry in the aerosol optical depth, but this is a relatively small effect compared with the asymmetry that is needed in the multi-objective strategy to compensate for the effect noted. A similar strategy executed in CESM1, which has similar stratosphere, but different cloud fast response to CO2, required more injection in NH to compensate (Fasullo and Richter, 2023).

We calculated the interhemispheric imbalance for zonal mean AOD, $l1$, for the hemispherically symmetric strategies 15N+15S, 30N+30S, and 60N+60S; the values of $l1$ for these three strategies are 0.004, -0.004, and -0.007, respectively, which are negligible compared to the magnitude of zonal mean AOD and much smaller than the value of $l1$ of -0.02 needed by the multi-objective strategy to compensate for T1. Thus, the hemispheric asymmetry in the multi-objective strategy is not due to BDC. We add a brief note to the text commenting on this.

Reference:

Fasullo, J. T. and Richter, J. H.: Dependence of strategic solar climate intervention on background scenario and model physics, Atmospheric Chemistry and Physics, 23, 163–182, https://doi.org/10.5194/acp-23-163-2023, 2023.

5.    *Page 9, line 221-222, this is misleading "This asymmetry arises as the northern hemisphere has a stronger Brewer-Dobson circulation than the southern hemisphere". The inter-annual variability, which is much larger in NH than in SH, is what causes the asymmetry as the BDC is stronger in SH than NH.*

We believe the statement in the current manuscript is correct; the stronger magnitude of climatological BDC in the NH than in the SH has been reported in a number of studies, e.g. Holton, 1990, Rosenlof and Holton, 1993, and Rosenlof, 1995.

List of references:

1.  Holton, J. R.: On the global exchange of mass between the stratosphere and troposphere, Journal of the Atmospheric Sciences, 47, 392-395, 10.1175/1520-0469(1990)047<0392:otgeom>2.0.co;2, 1990.
2.  Rosenlof, K. H., and Holton, J. R.: Estimates of the stratospheric residual circulation using the downward control principle, Journal of Geophysical Research-Atmospheres, 98, 10465-10479, 10.1029/93jd00392, 1993.
3.  Rosenlof, K. H.: Seasonal cycle of the residual mean meridional circulation in the stratosphere, Journal of Geophysical Research-Atmospheres, 100, 5173-5191, 10.1029/94jd03122, 1995.

6.    *Page 10, line 229, please add "the seasonality in" before "the Brewer-Dobson circulation" you add these citations.*

We made the change.

7.    *Page 10, line 229, please remove "also".*

We reworded the sentence on line 230 as follows: "The distribution of AOD in the annual injection cases exhibits a marked seasonal cycle, with extratropical AOD maximizing in winter and spring at each hemisphere, due to seasonality in the strength of the stratospheric transport".

*8.    Page 12, line 265, this "as solar reduction doesn't significantly change the Walker Circulation" is not clear. Please clarify or remove it.*

We modified the sentences to read:

"The pattern is similar to the pattern associated with the positive phase of the El-Nino Southern Oscillation (ENSO; e.g. McGregor et al., 2022), albeit differing in the strength and horizontal extent of the anomalous warming in the eastern equatorial Pacific. This is associated with changes in the strength and the position of the Walker Circulation (Bednarz et al., 2023-strategy2), caused likely in part by the the aerosol heating in the lower stratosphere (Simpson et al., 2019), and the resulting changes on tropical precipitation patterns in the region."

*9.    Page 12, line 275-279, this paragraph is not clear. Please rephrase it.*

We made a few edits on this paragraph and combined this paragraph with the previous paragraph. "However, we find that in all SAI strategies, the rms temperature change is larger than the rms temperature change that one would expect due to natural variability alone (i.e. 0.08∘ C, represented by the dashed line in Fig. 8a), indicating imperfect compensation of the pattern of warming under climate change. Among the SAI strategies considered here, the multi-objective strategy best minimizes the spatial rms of temperature changes, as indicated by the lowest rms temperature change (rms T=0.38∘ C). The 60N+60S strategy results in an uneven cooling with the highest rms temperature change (rms T=0.57∘ C), but still much smaller than the rms temperature change in SSP2-4.5 without SAI (rms T=1.53∘ C)."

*10.   The result about precipitation responses in section 4.3.2 are overlooked. Please better discuss these results.*

We have added further discussions on the regional precipitation responses in section 4.4 as described earlier.

*11.   Page 13, line 292-293, this "The difference in rms ... temperature responses." is not correct for Amazonia & congo basin (Fig 9).*

We have updated this sentence to "The difference in the spatial rms of P-E and precipitation responses between SAI simulations and the SSP 2-4.5 simulation is notably smaller than the difference in rms temperature responses, indicating poorer compensation of these metrics than temperature". The metrics rms temperature, rms P-E, and rms precipitation are global metrics (integrating regional changes across the globe), thus your statement is not appropriate here, but we do add further discussion of the regional changes over these regions as described earlier.

*12.  Page 14, line 295, this "the difference is no more than 30 %" is really misleading as the precipitation changes as well as their importance on mainland and certain key regions are not homogeneously distributed.*

We have modified the sentence on Line 295 as "For the corresponding changes in precipitation over the whole Earth surface (i.e. both land and ocean), the difference in rms precipitation response is no more than 30% when comparing any SAI strategy with the SSP2-4.5 case (Fig. 8(b)). While the difference in these global metrics between two strategies is modest, the corresponding regional P-E and precipitation changes can be locally statistically significant and thus need to be evaluated individually (Fig. 12)".

*13.  Page 14, line 308-311, A discussion about the link between TICZ changes with different SAI strategies is missing.*

We have added the following sentences in the paragraph on line 308-311:

Line 310: "The difference in the shift of ITCZ between the hemispherically-symmetric injection cases is modest, generally within one standard error."

Line 311: "The multi-objective strategy is the only one that explicitly targets hemispheric asymmetry; while T1 is an imperfect proxy for managing ITCZ, it does result in improved compensation relative to the hemispherically-symmetric strategies, indicating the value of including an objective associated with asymmetric compensation."

14.  Figures 8 and 10 should be combined for clear discussion and reduction the numbers. For instance global and regional plots separately.

We thank the reviewer for the comment. However, as also noted in our response to the major point #4, we don't think those two figures should be combined, as Figure 8 shows the rms changes of temperature and precipitation while Figure 10 shows ITCZ.

15.  There is needs for separating global and region response better from page11 to the end.

We thank the reviewer for the comment. As noted in our response to major point #3, we separate the discussions on global surface climate responses and regional surface climate responses into different sections.

16.  Page 17, line 326, it is not figure Fig 11a but Fig. 12a.

We thank the reviewer for this comment. We have corrected this in Line 326.

17.  Page 17, Paragraph 338-341 is speculative. Please rephrase it.

We have deleted this paragraph.

18. Please move most of the Appendix figures into the main text discussed.

We thank the reviewer for this comment. As noted in our response to major point #5, we have moved most figures in the appendix to the main paper. We have moved Figure A3-A5 to Supplementary materials.

---

## Referee Report (RR1)

**Comments on "Hemispherically-Symmetric Strategies for Stratospheric Aerosol Injection " by Zhang al.**

Zhang et al document the response of the climate system for a given stratospheric aerosol injection (SAI) strategy. To maintain the same mean surface temperature, the authors design several SAI strategies to be studied using the CESM2 Earth System Model (WACCM6-MA) and compare the results with the reference scenario, the SSP2-4.5 global warming scenario. My main concern with this paper is the lack of caution in the way the results are presented here as well as the lack of context on the complexity of our climate system and associated feedbacks, stratospheric circulation and variability as well as model biases in terms of precipitation and the lack of model representation of our complex system. I highlighted this point in my previous review, but it is not seriously addressed. The review does not address all my concerns, which is why I recommend major revisions. From a single model study, the authors intend to generalize their experiments to all models. The conclusion and abstract still marginalize the impact of SAI on precipitation and it even seems to me that the authors exaggerate their results by saying that SAI is good because it decreases temperature while ignoring the importance of the impact of SAI on precipitation and the related implication on food security, agriculture and so many others vital component for human survival. I am therefore going to reject this article and encourage for resubmission after addressing these serious issues.

**Major points:**
1. The authors did not address this major issue: "The surface climate response to different SAI strategies is present without a clear understanding of the impact of model internal and inter-annual variability on the distribution of SAI in the stratosphere as well as its feedback on the surface climate. According to Bittner et al. (2016), 7 ensembles in the tropics and 40 ensembles in the extra-tropics are needed to accurately capture the model circulation response to SAI, and hence the corresponding feedback on surface climate. Caution should be exercised in discussing the results here. Three ensembles are not enough to constrain the internal variability of the model". I invite them to red the Bittner et al 2016.
2. The abstract does reflect the content of the paper. Therefore, it needs to be rewritten

3. Page 1, line 20, please added after "latitudes" this "based on a single model study".

4. Introduction, please add a section on model limitation and biases regarding SAI and precipitation before "Defferent SAI strategie…" Models in general even struggle to reproduce Pinatubo or later volcanoes. Current climate response to SAI are still model dependent due to unconstrained internal model variability as well as interannual and decadal variability of the climate system.

5. Page 1, line 1 please added after "scenario" this "based on a single model study".

6. Page 3, line 80 please add after "variability" tis "… and model biases".

7. Page 4, line 92, the linearity hypotheses is not taking into account the feedback processes (Stratospheric water vapor, O3 and SAI) on circulation and climate.

8. Page 4, line 120, "significant perturbation of the interhemispheric temperature gradient and the associated location of tropical precipitation" only correspond to fast response of the SAI as the aerosol can be transported into deep stratosphere then impact the opposite hemisphere few month later.

9. Table1: How these aerosols and their life time are sensitive to the altitude of injection.

10. Page 9, line 223-224: This is due partly to the altitude of injection of SO2 as the shallow branch of the BDC will wash out the aerosols.

11. Page 9, line 227, I am still not convince that the existing BDC asymmetry between SH & NH due to the wave activitives, Polar vortex, internannual variability modulation has no impact on the aerosol distribution in the lower stratosphere (Fu & Qu, 2013).

12. Page 13, line 292, Please add after "climate." this "based on the WACCM results"

13. Page 24, line 477 after "injection strategy" please add " as well ad the models ability to capture the complexity of the Earth climate system and its variability, which is not investigated here."

14. Page 24, line 477 please "based on a single model study" after "SAI strategies …"

15. Page 24, line 481 please "based on a single model study" after "fundamental limits of SAI".

16. Page 25, line 487, what about precipitation results?

---

## Author Response (AR2)

We thank the editor and the reviewer for their careful reading. We address each of the points raised below. Our comments are in blue below, first to the editor's comments and then to the reviewer's.

I apologise for the delay in returning this manuscript to you. One of the two reviewers of the revised submission still raises concerns on the lack of contextualisation of the uncertainties/limits of the significance testing in your results. In your replies, you argued for the fact that large enough differences are robust even between individual model ensemble members. This may be true, but I don't find the authors' replies on this point to be entirely satisfactory. If one takes a single CMIP6 model run, for example, some measure of variability around the mean would likely result in a narrower range than the uncertainty deriving from running 30 or more model ensemble members, thus giving statistical overconfidence in your results. Another example of perhaps overconfident statistical testing, is the use of Welch's t-test. The authors never specify whether this assumes that the data being tested follows a specific underlying distribution, and never verify whether their data satisfies this. They also quote Wilks, but never explicitly mention whether a multiple testing correction is implemented (as per the Wilks 2016 "Stippling" paper).

I do not agree with the Reviewer that the above is sufficient grounds for rejection, but it is definitely something that should be addressed with a combination of careful rewording - to acknowledge that there are, like in all studies, some caveats to the conclusions being drawn - and reconsideration or further justification of some of the adopted statistical testing approaches. I also ask the authors to provide point-by-point replies to the rest of the remaining Reviewer comments.

We agree with the editor on the importance of including appropriate caveats on use of statistics and have added several clarifying comments and cautions. The reviewer's primary concerns are around model error in relying on a single climate model, and we have added further emphasis on this caveat in the abstract and conclusions in particular.

Regarding the editor's concerns on statistics:

(1) First, regarding the number of ensemble members, it is certainly true that when one has very few data points one does not obtain an adequate estimate of the standard deviation, leading to overconfidence (or sometimes underconfidence). Here, we have 3 ensemble members and look at 20 years of simulation output for each variable, so we have a total of 60 data points. Accounting for autocorrelation in the time series results in fewer independent data points, with the exact number dependent on the specific variable being considered; we believe that this is sufficient to estimate the standard deviation (which is then used by the t-test to assess whether differences are statistically significant). Note that while we disagree with the reviewer on the ability to define a single universal number of data points for all

statistical testing in climate science (simply because that number clearly depends on the signal, so the size of the forcing and how that forcing affects a particular variable, and also on the variability of the variable in question), the number of independent data points here is indeed larger than the estimate from Bittner et al that the reviewer suggests.

(2) Second, a t-test assumes that normality is a reasonable approximation for the underlying probability distribution; this is a good assumption for annual-average outputs from climate models (due to the central limit theorem). We add a note to this effect.

(3) Third, we did not use a multiple testing correction but we have added a caution in several places regarding the fact that we test many different variables and that as a result it is possible that some differences may appear to be statistically significant simply by chance. In particular, at the beginning of the results section where we describe the statistical testing used herein, we add "We use t-tests to estimate significance, which assume that variability is approximately normal; this is a reasonable approximation for annual-mean climate variables. One caution on interpreting results is that in evaluating many different climate variables, some will appear to be statistically significant at the 95% level by random chance (Wilks, 2016)." We add a similar comment at the end of the Summary section.

Comments on "Hemispherically-Symmetric Strategies for Stratospheric Aerosol Injection" by Zhang al.

Zhang et al document the response of the climate system for a given stratospheric aerosol injection (SAI) strategy. To maintain the same mean surface temperature, the authors design several SAI strategies to be studied using the CESM2 Earth System Model (WACCM6-MA) and compare the results with the reference scenario, the SSP2-4.5 global warming scenario. My main concern with this paper is the lack of caution in the way the results are presented here as well as the lack of context on the complexity of our climate system and associated feedbacks, stratospheric circulation and variability as well as model biases in terms of precipitation and the lack of model representation of our complex system. I highlighted this point in my previous review, but it is not seriously addressed. The review does not address all my concerns, which is why I recommend major revisions. From a single model study, the authors intend to generalize their experiments to all models. The conclusion and abstract still marginalize the impact of SAI on precipitation and it even seems to me that the authors exaggerate their results by saying that SAI is good because it decreases temperature while ignoring the importance of the impact of SAI on precipitation and the related implication on food security, agriculture and so many others vital component for human survival. I am therefore going to reject this article and encourage for resubmission after addressing these serious issues.

While we did include caution regarding the broader interpretation from a single-model study, we have made this caveat more explicit both in the abstract and in the introduction, where we add a sentence about using these results to motivate model intercomparisons. The final paragraph of the conclusion section already emphasizes that the results are obtained in a single model and that multi-model explorations are critical; the penultimate conclusion paragraph further articulates that one motivation for the current study is to support GeoMIP model intercomparisons. We thus feel that this point is already well emphasized in the conclusions, though we add a further note about the need for multi-model studies in the first paragraph of the conclusions as well.

Note, of course, that the same concern regarding use of a single model could be made regarding the majority of papers in climate science. At no point do we claim that all models would behave the same way as this one, and in fact one of the points of our study is to introduce a common framework to use in future model comparisons in which inter-model biases and differences can be assessed explicitly (noted in the conclusions). In addition, one of the motivations for considering only hemispherically-symmetric injection strategies is that the asymmetry required to compensate for interhemispheric asymmetry in temperature is expected to be model-dependent, as we note on line 105 of the previous version.

Further, the paper never says that SAI is "good" (or similar normative statements) but simply objectively reports findings on temperature, precipitation and other climate metrics. Further analysis of the associated changes in stratospheric and tropospheric temperatures, circulation and chemistry, and their links to the surface climate responses in these simulations is made in our other recently published study (Bednarz et al., 2023), to which we add a citation at the end of the introduction. We also note that in response to the reviewer's previous review comments, we greatly expanded the discussion of regional precipitation responses including additional figures on the differences in response over the Congo and Amazon basins; as noted previously we thank the reviewer for pointing out that insufficient emphasis in our original submission.

Major points:
1. The authors did not address this major issue: "The surface climate response to different SAI strategies is present without a clear understanding of the impact of model internal and inter-annual variability on the distribution of SAI in the stratosphere as well as its feedback on the surface climate. According to Bittner et al. (2016), 7 ensembles in the tropics and 40 ensembles in the extra-tropics are needed to accurately capture the model circulation response to SAI, and hence the corresponding feedback on surface climate. Caution should be exercised in discussing the results here. Three ensembles are not enough to constrain the internal variability of the model". I invite them to read the Bittner et al 2016.

We disagree with the appropriateness of choosing a single universal number for the "correct" number of ensembles and the relevance of the specific quantification from Bittner et al to our work. Bittner et al. (2016) shows that a large number of ensemble members are required to detect NH polar vortex strengthening in the first post-Pinatubo winter. This number depends on

the latitude considered and ranges from 7 at the southward flank of the maximum positive wind anomaly to more than 40 members at high latitudes." (Section 4 of Bittner et al.).

First, our current study does not deal with the changes in the NH polar vortex, the variability of which can indeed be particularly large (we note that this aspect of the climate response to SAI in these simulations has been assessed in our other paper, Bednarz et al., 2023, which has now undergone a thorough peer review and is published).

Second, our study does not analyze changes in a single year or a single season as done in Bittner et al. study - in fact we analyze changes over the means over the 20-year long period and 3 ensemble members, and so we have effectively 60 semi-independent data points, which is indeed larger than the number the reviewer argues is necessary. (The effective degrees of freedom need to be adjusted to account for autocorrelation in the time series, but this still leads to more than the 40 independent degrees of freedom noted by the reviewer for every variable except global mean temperature.)

Third, the ability to distinguish a response depends on the magnitude and duration of the perturbation, and as such there are large differences between a short-term volcanic forcing and a continuous SAI.

Lastly, the reviewer is correct in that more ensemble members gives better ability to distinguish signal from variability. However, there is no single threshold that can be defined for the required number of ensemble members in general, and thus generalizing the single model and single-forcing study by Bittner et al. to all models and all forcing scenarios is not appropriate.

The effects of internal variability are already included in all analyses presented, all of which include error bars indicating the limit of confidence resulting from variability, as is standard practice. We account for autocorrelation in the time series. We have also added a comment in discussing statistical testing in general about the assumption of normality embedded in using a t-test (which is a very good assumption, following from the central limit theorem), and about the caution in evaluating statistical significance for many different variables that some may appear to be significant by chance; this latter comment we also reiterate at the end of the Summary section.

2. The abstract does reflect the content of the paper. Therefore, it needs to be rewritten

We respectfully disagree with the reviewer. If there are any specific aspects of the abstract that do not reflect the content of the paper, we would appreciate it if those could be pointed out.

3. Page 1, line 20, please added after "latitudes" this "based on a single model study".

To improve the emphasis in the abstract, we add "in one climate model" earlier in the abstract on line 7. On line 20 this is already abundantly clear from context as the sentence is describing

the results in the paper.  (Nonetheless, the sentence noted here should reasonably be expected to hold in any climate model.)

4. Introduction, please add a section on model limitation and biases regarding SAI and precipitation before "Defferent SAI strategie…" Models in general even struggle to reproduce Pinatubo or later volcanoes. Current climate response to SAI are still model dependent due to unconstrained internal model variability as well as interannual and decadal variability of the climate system.

CESM(WACCM) does a reasonable job of matching stratospheric aerosol distribution after Pinatubo, as was investigated by Mills et al. (2017); we have now added a sentence noting this, in Section 2 where the climate model used herein is described.  There are multiple examples over the past several decades of models actually reproducing quite well both Pinatubo and later volcanoes - both in terms of the aerosol distribution, and in terms of the overall climate response.  Of course we agree that any model projection should be interpreted as the predictions of a model.

5. Page 1, line 1 please added after "scenario" this "based on a single model Study".

The word "scenario" does not show up on page 1, line 1, and we are not clear what line the reviewer intended to refer to.  We do add "in one climate model" on line 7 to be explicit earlier in the abstract.

6. Page 3, line 80 please add after "variability" tis "… and model biases".

The suggested change would not be correct. The sentence refers to the ability to detect changes, which is limited by internal variability but not by model bias.  Model bias is a source of uncertainty (and thus the differences found in one model may not be the same as what would happen in the real world).  Errors in the model's ability to represent natural variability accurately would also lead to an error in estimating detectability; factors like that are why the sentence already says "among other factors".

7. Page 4, line 92, the linearity hypotheses is not taking into account the feedback processes (Stratospheric water vapor, O3 and SAI) on circulation and climate.

The sentence is true as written.  The presence of feedback is irrelevant to whether linearity is adequate. What is relevant is whether the feedback is itself significantly nonlinear.
There are ample examples of feedbacks in linear systems (see, e.g., Astrom and Murray, https://fbswiki.org/wiki/index.php/Feedback_Systems:_An_Introduction_for_Scientists_and_Engineers  or any other textbook on feedback systems).  Ultimately the only question is whether or not linearity is an adequately good approximation as there is no physical system as a truly linear system.  The adequacy of the assumption would need to be validated in this particular context, which is why it is noted in passing here simply as a possible choice of assumption in future work, but there is ample support for linearity being an adequate approximation for SAI in

particular (e.g., MacMartin et al. 2013, Irvine et al. 2019, MacMartin et al. 2019, Visioni et al. 2023, etc.) and so it is not unreasonable to expect that it will likely prove sufficient.

8. Page 4, line 120, "significant perturbation of the interhemispheric temperature gradient and the associated location of tropical precipitation" only correspond to fast response of the SAI as the aerosol can be transported into deep stratosphere then impact the opposite hemisphere few month later.

The reviewer's claim is not correct. Injecting away from the tropics in a single hemisphere will put aerosols primarily in that particular hemisphere (constrained by Brewer-Dobson circulation), and indeed lead to an asymmetric aerosol burden and corresponding forcing, and a resulting shift in tropical precipitation.  This is well documented in numerous papers including the one cited here (Haywood et al. 2013).  We therefore retain the current wording.

9. Table1: How these aerosols and their life time are sensitive to the altitude of injection.

The reviewer is correct that the results will also be somewhat sensitive to the altitude of injection, although as long as the injection is not too close to the tropopause the main effect of altitude will be to change the injection rates required to achieve a given cooling and not the spatial pattern of that cooling (Lee et al. 2023); we now add a sentence describing the choice of altitude and the effect of that choice.

10. Page 9, line 223-224: This is due partly to the altitude of injection of SO2 as the shallow branch of the BDC will wash out the aerosols.

True that the altitude affects lifetime; the relevance of altitude is now noted earlier as commented on above.

11. Page 9, line 227, I am still not convince that the existing BDC asymmetry between SH & NH due to the wave activitives, Polar vortex, internannual variability modulation has no impact on the aerosol distribution in the lower stratosphere (Fu & Qu, 2013).

We are confused about this comment; of course BDC asymmetry affects aerosol distribution and we never claimed that it didn't.  But that is a small effect compared to the asymmetry arising from injecting 12.2 Tg/yr in one hemisphere and 5.8 Tg/yr in the other, which is what the line in question is pointing out.

12. Page 13, line 292, Please add after "climate." this "based on the WACCM results"

The sentence in question is describing Figure 8; we believe that it is clear that Figure 8 is presenting results from the simulations described herein and that it is unnecessary to repeat "based on the WACCM results" in this and any other sentence in the paper that is describing simulation results.  The reviewer's suggestion would be appropriate in conclusions that could reasonably be inferred as making claims broader than these simulations.

13.Page 24, line 477 after "injection strategy" please add " as well ad the models ability to capture the complexity of the Earth climate system and its variability, which is not investigated here."

The sentence in question is accurate as written.  That is, that we have to be able to assess the role of scenario and strategy as part of assessing possible outcomes of SAI.  It is also true that we need good enough models, but that is not the focus of this paper - there are many papers on that subject already, while the purpose herein is to highlight the importance of the strategy dimension in particular.  Much of the rest of the conclusion section already emphasizes the limitations of a single-model study and the need to conduct analyses in multiple models.

14.Page 24, line 477 please "based on a single model study" after "SAI strategies …"

We think the reviewer was intending to refer to line 478 (as "SAI strategies" does not appear in Line 477).  The sentence in question simply notes that different SAI strategies will affect the surface climate differently; that is clearly a conclusion that would hold in any climate model as well as in the real world.  Rather than adding a comment both here and in the next sentence as the reviewer suggests, we include an additional comment towards the end of the paragraph noting the need to conduct similar analyses in multiple models; this provides emphasis that the results are based on a single model.  This point is now made in every paragraph of the Discussion section.  We furthermore add this point in two places in the Summary section describing the results.

15. Page 24, line 481 please "based on a single model study" after "fundamental limits of SAI".

We add an additional note about the importance of conducting similar analyses several sentences later "as well as to conduct similar analyses in other climate models".

16. Page 25, line 487, what about precipitation results?

The previous paragraph notes that there are many climate goals that are relevant and could be optimized over, not just temperature or precipitation.

---

## Author Response (AR3)

We thank the editor for their comments. Our response is in blue below.

1. Concerning the number of ensemble members, it would make for an interesting discussion point to mention the paper by Bittner et al. (2016) and explain – as you do in your replies – why those conclusions do not necessarily apply to your work. This can be very short, but may help attentive readers to better understand the rationale behind the robustness of your results.

Towards the end of the Simulation section, We added " The surface climate responses are evaluated based on the 20-year average over the period of 2050-2069. With three ensemble members, the 20-year average of each evaluated climate variable is calculated based on 60 annual-mean values. Taking into account temporal autocorrelation, the effective sample size is still comparable to the suggested number of independent data points (20-40) in Pausata et al. (2015). This effective sample size is also comparable to the suggested sample size (7-40) in another relevant study focusing on discerning NH polar vortex change from internal variability (Bittner et al., 2016). As this study focuses on the long-term impacts of continuous injection, rather than impacts of a pulse volcanic eruption in the single year following the eruption (as in, e.g., Pausata et al., 2015; Bittner et al., 2016), data from three ensemble members are likely sufficient to distinguish a signal over a 20-year period from internal variability. "

2. In the abstract, you could consider removing or shortening the following passage: "therefore, understanding the range of possible climate outcomes is crucial to making informed future decisions on SAI, along with the consideration of other factors. Yet to date, there has been no systematic exploration of a broad range of SAI strategies. This limits the ability to determine which effects are robust across different strategies and which depend on specific injection choices, or to determine if there are underlying trade-offs between different climate goals." This reads more like a passage from the introduction, justifying the novelty of the article, than an abstract. I however accept that this point is to some extent subjective.

We have shortened this passage by removing the following part, ", or to determine if there are underlying trade-offs between different climate goals".

3. As Wilks (2016) points out, multiple testing issues can lead to spurious spatial patterns of significance, which are impossible to evaluate a priori and which can in turn lead to incorrect interpretation of results. Since you cite Wilks and are clearly aware of the issue, simply mentioning it in a sentence but not correcting your figures is hard to justify scientifically. This is particularly relevant for the global geographical plots. I would warmly encourage you to update your testing for these. There are ready-to-use packages performing multiple testing correction in several programming languages, which cause only a modest computational overhead.

We thank the editor for their comment. We have performed multiple testing on regional changes in temperature, precipitation, and P-E to account for spatial correlation, and updated the t-test results in Figures 8-10 in the manuscript and Figures S2-S6 in the Supplement. In Line 168 in the manuscript, we added "We also perform multiple testing to account for spatial correlation

using the false discovery rate (FDR) method, where we choose $\alpha_{FDR}=0.1$ for achieving a global significance level of 0.05 based on the conclusion in Wilks (2016)."